# Universal Structures and Emergent Geometry from Large-$c$ BCFT Ensemble

**Ling-Yan Hung,**[a,b] **Yikun Jiang,**[c] **Bingxin Lao**[d]

[a] *Yau Mathematical Sciences Center, Tsinghua University, Beijing 100084, China*

[b] *Yanqi Lake Beijing Institute of Mathematical Sciences and Applications (BIMSA), Huairou District, Beijing 101408, China*

[c] *Department of Physics, Northeastern University, Boston, MA 02115, USA*

[d] *Department of Physics, Princeton University, Princeton, NJ 08544, USA*

E-mail: elektron.janethung@gmail.com, phys.yk.jiang@gmail.com, bingxin.lao@princeton.edu

ABSTRACT: In this paper, we study the ensemble average of boundary CFT (BCFT) data consistent with the bootstrap equations. We apply the results to computing ensemble average of copies of multi-point correlation functions of boundary changing operators (BCO), and find the results in agreement with one copy of the Virasoro TQFT. Further, we consider ensemble average of CFT path-integrals expressed as tensor networks of BCO correlation functions using the formalism developed in [1–3]. We find a natural emergence of locality and a loop-sum structure reminiscent of lattice integrable models. We illustrate this universal structure through explicit examples at genus zero and genus one. Moreover, we provide strong evidence that, at leading order in large-$c$, the results match those of three-dimensional Einstein gravity. In the presence of closed CFT operator insertions, generalized free fields emerge, with their correlation functions governed by the shortest paths connecting the insertions.

## 1  Introduction

Quantum chaotic systems are known to exhibit universal behaviors. The eigenstate thermalisation hypothesis (ETH) suggests a remarkably general universality of expectation values of simple operators in highly excited states [4, 5]. This, and its generalisations thereof has been considered in the context of 2D conformal field theory (CFT) [6], leading to constraints and predictions for the universal behavior of operator product expansion (OPE) coefficients for heavy states in CFTs [7], and further refined in the context of CFTs with large central charge (large-$c$) [8, 9].

Meanwhile, the inclusion of wormhole geometries in the semi-classical sum over bulk saddles strongly suggests that a generic gravitational path-integral could be dual to an ensemble of CFTs [10, 11]. There is growing evidence that the semi-classical limit of gravity should be viewed as a coarse-grained description of its CFT dual, and simple observables, such as few-point functions, may not be sensitive to all microscopic details of the theory. This motivates the study and construction of ensembles of large-$c$ CFTs that realize ETH-like behavior and reproduce features of semi-classical gravitational duals. A natural approach is to treat OPE coefficients and conformal dimensions as random variables [8, 9, 12, 13], and to construct distributions that reproduce universal features such as ETH. Recent works have shown that suitable Gaussian random matrix models for the CFT data can match semi-classical gravitational results for various correlation functions. It is proposed that one can treat the OPE coefficients and conformal dimensions of operators as random variables [8, 9, 12, 13], and construct distributions that reproduce universal behaviour such as the ETH.

Each individual CFT is expected to obey local consistency conditions such as crossing symmetry. However, it has been proposed that semi-classical gravity may be insensitive to small violations of these constraints, suggesting that the ensemble of CFTs dual to gravity could admit slight deviations. Accordingly, one can relax these constraints and impose them perturbatively, order by order in an expansion in the large-$c$. An explicit construction of such an ensemble was proposed in [12, 13], where the ensemble average extends beyond the Gaussian approximation in [7]. Remarkably, this approach reproduces a sum over bulk geometries corresponding to different three-manifolds, in agreement with the structure of the Virasoro TQFT [14].

In a complementary direction, in [1, 2], we proposed a discretization of CFT path integrals by decomposing smooth rational CFT (RCFT) path integrals into products of open pairs of pants—specifically, three-point functions of boundary-changing operators (BCOs). We further showed in [3] that this framework extends to the irrational Liouville theory, indicating that it may be applicable more generally to large-$c$ holographic CFTs.[1]

It is thus very tempting to extend the discussion in [8, 9, 12, 13] to the open sector of CFTs, and to model the structure coefficients of BCOs as random variables. Boundary CFT (BCFT) also exhibits universal behaviour of heavy states [15, 16], providing additional motivation. Inspired by these developments, we propose a generalization of the random tensor models for BCO structure coefficients, which are approximately Gaussian at leading order, with crossing relations imposed order by order in $1/c$.

In this paper we first propose a distribution for the OPE coefficients involving the BCOs, and discuss non-Gaussianities analogous to discussion in [12, 13, 17, 18], that restore crossing symmetry at subleading orders. We then consider averages of products of BCO correlation functions, and find a precise connection between this averaged structure and a single copy of the Virasoro TQFT proposed in [14].

Next, we apply this framework to compute CFT path integrals over various 2D mani-

---

[1]We emphasize that Liouville theory is not a holographic 2D CFT in the conventional sense; see [3] for detailed discussion.

folds—each decomposed into a product of BCO correlation functions as described above—and then take the ensemble average. Remarkably, this computation reduces to a sum over loop configurations, reminiscent of those appearing in well-known lattice integrable models [19, 20]. We further generalize our analysis to include operator insertions in the 2D path integral. In this case, geodesic lines connecting the inserted operators dominate in the presence of a background loop ensemble, leading to the emergence of generalized free field behavior. As we will show, the emergence of bulk locality and generalized free fields appears to be a universal feature of large-$c$ CFTs. We provide evidence that is matches with the usual 3d pure gravity results.

We illustrate the computation of path integrals on genus-zero and genus-one surfaces as two explicit examples, obtaining universal results.

Our paper is organised as follows. In Section 2 we will start with a review of relevant results on the ensemble average of closed CFT structure coefficients. Then we will generalise our discussions to include ensemble average of BCFT data, and discuss the implementation of crossing symmetry in the open sector. In Section 3, we compute ensemble averages of products of BCO correlation functions and establish their connection to the Virasoro TQFT. In Section 4, we apply this framework to study triangulation of CFT path-integrals, expressed in terms of BCO correlation functions on manifolds of various topologies. We also discuss average of few point closed correlation functions expressed in terms of averages of BCO and bulk boundary couplings, and demonstrates the emergence of generalised free fields. We conclude with a summary and outlook in Section 5. Reviews of useful prior results, and our conventions and normalisations, are provided in the appendices.

At the final stage in the preparation of the current manuscript, we are made aware of the new paper [21], which has some overlap with our current paper on the ensemble average of BCFT data. Our discussion here however focuses on the alternative perspective of ensemble average of triangulated smooth path-integrals and their implications.

## 2 Ensemble average of CFT and BCFT

In this section we discuss generalising ensemble average of large-$c$ closed CFT data to include BCFT data in the average. We first review the asymptotic behaviour of large-$c$ CFTs and BCFT. We will make some natural assumptions about our ensemble, similar to the approach in [9, 12]. Then we introduce the Gaussian variance for the OPE coefficients. Analogous to the crossing symmetry violation problem in bulk CFT, which is carefully examined in Appendix A.3, we need to introduce non Gaussianity to restore the crossing symmetry. To avoid clutter, we also include a review of bulk CFT and BCFTs with some necessary but well-known details in Appendix A, ensuring completeness without interrupting the flow.

## 2.1 Asymptotic behaviour of heavy states in closed CFTs

In this paper, we consider an ensemble of generic large-$c$ CFTs, each potentially with a different spectrum.[2] The spectra of irrational large-$c$ CFTs can be generically divided into two regimes. It involves a regime of heavy states above the black hole threshold, in which the conformal dimension $h \geq \frac{c-1}{24}$. Virasoro representations form a continuous collection in this regime. There is also a regime below threshold where Virasoro representations come at discrete points for $0 \leq h < \frac{c-1}{24}$.

### Liouville Parametrization for 2D CFTs

There is a convenient parametrisation of Virasoro representations in terms of the Liouville coupling $b$ for central charge $c > 1$,

$$c = 1 + 6\left(b + \frac{1}{b}\right)^2 = 1 + 6Q^2, \quad 0 < b < 1, \tag{2.1}$$

Conformal dimensions can be labeled by the Liouville momentum $P$, or equivalently a complex number $\alpha := \frac{Q}{2} + iP$ of which the conformal dimension is

$$h = \alpha(Q - \alpha) = \frac{Q^2}{4} + P^2. \tag{2.2}$$

The heavy states mentioned above correspond to $P \geq 0$.[3]

### Universal Cardy density of states

The spectrum of the closed CFT can be encoded in the torus path integral $Z_{\mathrm{CFT}}(\tau, \bar{\tau})$, where $\tau$ is the torus modulus. This partition function can be decomposed in terms of Virasoro characters:

$$Z_{\mathrm{CFT}}(\tau, \bar{\tau}) = \int dP_L dP_R \rho(P_L, P_R) \chi_{P_L}(\tau) \chi_{P_R}(\bar{\tau}). \tag{2.3}$$

where $\rho(P_L, P_R)$ denotes the density of states, which, for CFTs with a discrete spectrum, consists of a sum over delta functions localized on the spectrum.

Under the high-temperature limit corresponding to $\tau \to 0$, the vacuum character $\chi_{\mathbb{1}}(-1/\tau)$ dominates in the dual channel for a unitary CFT with a gap above the vacuum. This yields a universal expression for the density of heavy states $h \gg c$, known as the Cardy formula [22],

$$\rho(P_L, P_R) \to \rho_0^{\mathrm{Cardy}}(P_L) \rho_0^{\mathrm{Cardy}}(P_R), \qquad \rho_0(P) = \mathbb{S}_{\mathbb{1},P}[\mathbb{1}], \tag{2.4}$$

where $\mathbb{S}_{\mathbb{1},P} = 4\sqrt{2} \sinh(2\pi bP) \sinh\left(\frac{2\pi P}{b}\right)$ as reviewed in the appendix.

---

[2]An alternative interpretation is that the ensemble arises from coarse-graining over energy windows within a single theory. Since the resulting physics is expected to be universal for large-$c$ holographic CFTs, we will not distinguish between these two perspectives in this paper.

[3]The $P < 0$ states are not independent from $P > 0$.

The density of states can also be expressed in terms of conformal dimensions $h$, which can be converted from the above expression in terms of $P$ to its more familiar form

$$\rho_0^{\text{Cardy}}(h)dh = \rho_0^{\text{Cardy}}(P)dP, \qquad \rho_0^{\text{Cardy}}(h) \to e^{2\pi\sqrt{\frac{c}{6}\left(h-\frac{c}{24}\right)}}. \tag{2.5}$$

**Asymptotics of OPEs**

By similar considerations, it is shown that the structure coefficients between three primary operators to take the asymptotic value [7]

$$C_{ijk}C_{ijk}^* \sim C_0(P_{L\,i}, P_{L\,j}, P_{L\,k})C_0(P_{R\,i}, P_{R\,j}, P_{R\,k}), \tag{2.6}$$

when at least one of the primary operators gets heavy. The normalisation conventions for the structure coefficients are reviewed in the appendix, and $C_0$ is given in (A.13).

## 2.2 Asymptotic behaviour of heavy states in BCFTs

Here we provide a brief review of asymptotic behaviour of heavy states in BCFTs. In addition to bulk CFT operators, there are boundary changing operators (BCOs) living on the boundary. We use $\Psi_i^{\alpha\beta}$ to denote the $i$-th primary BCO, inserted between the boundaries labeled by $\alpha$ and $\beta$. In this paper we always denote the boundary conditions with Greek letters $\alpha, \beta, \gamma \ldots$. Primary operators are labeled by alphabet $i, j, k, \ldots$. The two point function for BCOs inserted on the real line is given by

$$\langle \Psi_i^{\alpha\beta}(x_1)\Psi_j^{\beta\alpha}(x_2)\rangle = \frac{g_i^{\alpha\beta}\delta_{ij}}{|x_{12}|^{2h_i}}, \tag{2.7}$$

where $x_{ij} := x_i - x_j$, and $g_i^{\alpha\beta}$ satisfies $g_i^{\alpha\beta} = g_i^{\beta\alpha}$. The three point function of these operators are[4]

$$\langle \Psi_k^{\alpha\beta}(x_1)\Psi_i^{\beta\gamma}(x_2)\Psi_j^{\gamma\alpha}(x_3)\rangle = \frac{C_{ijk}^{\alpha\beta\gamma}}{|x_{12}|^{h_{ki,j}}|x_{23}|^{h_{jk,i}}|x_{ki}|^{h_{ki,j}}}, \tag{2.8}$$

The overall constant $C_{ijk}^{\alpha\beta\gamma}$ is called the (open) structure coefficient of 3-point BCOs. It is easy to see that the structure coefficients respect cyclic symmetry

$$C_{ijk}^{\alpha\beta\gamma} = C_{jki}^{\beta\gamma\alpha}, \qquad \text{(triangle diagram)} . \tag{2.9}$$

which can be graphically represented as the rotational symmetry of the triangle (see the definition of the triangle in (B.11)). In the following diagrams, we always use red to denote

---

[4]Notice that the structure coefficient is defined to be the coefficient appearing in the three point function, and it is different from the OPE coefficients up to the normalization factor in the two point function (2.7). We also adopt a slightly unusual ordering convention for the labels in the structure coefficients, which is chosen to more closely match our corresponding diagrammatic notations.

the conformal boundary, blue to represent (states correponding to) BCOs, and black for the conformal blocks, which is defined on the dual graph for the blue lines. A more detailed discussion of our convention can be found near (B.9) in the appendix. Notice that two point functions can be obtained through three point functions by setting one operator to be $\mathbb{1}$ and identify boundary condition properly, we have

$$g_i^{\alpha\beta} = C_{i\mathbb{1}i}^{\alpha\beta\alpha}. \tag{2.10}$$

We have fixed the normalisation for bulk operators as $\langle \mathcal{O}_i(0)\mathcal{O}_j(1) \rangle = 1$ so we are not completely free to choose the normalisation for BCOs. For convenience we adopt the same convention as in [16], setting

$$g_i^{\alpha\beta} = C_{i\mathbb{1}i}^{\alpha\beta\alpha} = \sqrt{g_\alpha g_\beta}, \tag{2.11}$$

where $g_\alpha$ denotes the disk partition function with boundary condition $\alpha$, also referred to as the $g$-factor, as introduced in (A.23).

**Density of states in the open Hilbert space**

For the BCO, there is an analogous universal open density of states, derived from open-closed duality [15, 16]. The path-integral $Z_{\mathrm{cyl}(\alpha\beta)}(\tau)$ on the cylinder with conformal boundary conditions labeled $\alpha$ and $\beta$ respectively on the top and bottom boundary, is related to the open density of states $\rho_{\alpha\beta}(P)$ via

$$Z_{\mathrm{cyl}(\alpha\beta)}(\tau) = \sum_P \rho_{\alpha\beta}^{\mathrm{open}}(P)\chi_P(\tau). \tag{2.12}$$

As reviewed in the appendix (A.37), the Cardy's condition implies

$$\rho_{\alpha\beta}^{\mathrm{open}}(P) = g_\alpha g_\beta \mathbb{S}_{\mathbb{1},P}[\mathbb{1}] + \sum_{\substack{\mathcal{O}_i \in \mathcal{H}_{\mathrm{closed}}^{\mathrm{sc.}} \\ \mathcal{O}_i \neq \mathbb{1}}} \mathcal{B}_\alpha^i \mathcal{B}_\beta^i \mathbb{S}_{P_i,P}[\mathbb{1}]. \tag{2.13}$$

Using (A.22) and (A.20), one sees that

$$\frac{\mathbb{S}_{P_i,P}[\mathbb{1}]}{\mathbb{S}_{\mathbb{1},P}[\mathbb{1}]} \sim \begin{cases} e^{-4\pi\alpha_i P} & \alpha_i = \frac{Q}{2} + iP_i \in \left(0, \frac{Q}{2}\right) \\ 2\cos(4\pi PP_i)e^{-2\pi QP} & P_i \in (0, \infty) \end{cases} \tag{2.14}$$

in the limit where the operator is heavy, i.e., $P \to \infty$. The second term in (2.13) is exponentially suppressed compared to the first leading term for fixed $P_i$. The analogue of Cardy's formula for density of states of the open Hilbert space is thus given by

$$\rho_{\alpha\beta}^{\mathrm{open}}(P) \sim g_\alpha g_\beta \rho_0(P), \quad P \to \infty. \tag{2.15}$$

**Asymptotics of BCO OPEs and bulk-boundary couplings**

Similar to the OPE coefficients $C_{ijk}$ of bulk operators, the boundary structure coefficient $C_{ijk}^{\alpha\beta\gamma}$ exhibits asymptotic behavior, when at least one of the operators is heavy [15, 16]. It

is given by

$$\left|C_{ijk}^{\alpha\beta\gamma}\right|^2 := C_{ijk}^{\alpha\beta\gamma}C_{kji}^{\gamma\beta\alpha} \sim C_0(P_i, P_j, P_k). \tag{2.16}$$

In addition, near the boundary $\alpha$, a bulk operator $\mathcal{O}_i(z)$ with conformal dimensions $(h_i, \bar{h}_i)$ can fuse with boundary operators, leading to a bulk-boundary two-point function,

$$\langle \mathcal{O}_i(z)\Psi_j^{\alpha\alpha}(x)\rangle = \frac{C_{ij}^{\alpha}}{|z-\bar{z}|^{2h_i-h_j}|z-x|^{2h_j}}, \tag{2.17}$$

where $h_j$ is the conformal dimension of $\Psi_j^{\alpha\alpha}$. One can also consider inserting both bulk operators $\mathcal{O}_i$ in the presence of conformal boundary $\alpha$ and BCO $\Psi_j^{\alpha\alpha}$. Analogously, there exists asymptotic behavior for the bulk-to-boundary coefficient when one of the operators become heavy [15, 16]

$$\left|C_{ij}^{\alpha}\right|^2 \sim C_0(P_i, \bar{P}_i, P_j) \quad P_i, \bar{P}_i \to \infty \text{ or } P_j \to \infty, \tag{2.18}$$

## 2.3 Ensemble average of CFTs and BCFTs

In holographic CFTs, which exhibit a large gap and a sparse spectrum below threshold, the validity of the above results for heavy states extends further [23]. For example, the density of states in (2.1) can be approximated by a continuous integral over the entire range of heavy states, in other words,

$$\rho(P_L, P_R)dP_L dP_R \approx \rho_0^{\text{Cardy}}(P_L)\rho_0^{\text{Cardy}}(P_R)dP_L dP_R, \quad P_L, P_R > 0. \tag{2.19}$$

This reflects a universal behavior of 2D holographic CFTs in line with the ETH. In [8, 9], it was proposed to treat the OPE coefficients as random variables, with variances governed by the universal expression in (2.6). Together with the approximate density of states, this statistical data defines an ensemble of large-$c$ CFTs.

As briefly reviewed above, both the closed CFTs and open CFTs exhibit universal asymptotic behaviors in the large-$c$ limit [7]. It suggests that one could consider an ensemble of BCFTs, much like the case of closed CFT [9].

### 2.3.1 Ensemble average of closed CFT

In [12, 13], it was proposed that one can systematically construct the statistical moments of the random OPE coefficients within this ensemble by leveraging the consistency conditions of two-dimensional CFTs—commonly referred to as bootstrap constraints—such as modular invariance and crossing symmetry. A key idea in these constructions is to allow the structure constants to fluctuate slightly away from exact satisfaction of the bootstrap equations, which are treated as effective potentials. As a result, the constraints are only approximately satisfied, with corrections organized in an expansion in $1/c$.

To extend our discussion to BCFT, let us first review the ensemble average in closed CFT.

First, in [9], it is assumed that $\overline{C_{ijk}} = 0$. Overlines are used to denote taking averages. The leading non-vanishing contribution arises from the Gaussian average of pairs of closed

structure coefficients, given by

$$\overline{C_{ijk}C^*_{lmn}} = C_0(h_i, h_j, h_k)C_0(\bar{h}_i, \bar{h}_j, \bar{h}_k)\left[\delta_{il}\delta_{jm}\delta_{kn} + \text{signed permutation}\right]$$
$$+ \left[\delta_{ij}\delta_{lm}\delta_{k\mathbb{1}}\delta_{n\mathbb{1}} + \text{signed permutation}\right], \tag{2.20}$$

where

$$C^*_{lmn} = (-1)^{J_l+J_m+J_n}C_{lmn} = C_{nml} \tag{2.21}$$

with $J_i$ denoting the spin of operator $\mathcal{O}_i$. The "signed permutation" means those terms come from the permutations of indices $l, m, n$ and one should include a sign $(-1)^{J_i+J_j+J_k}$ if the permutation is the odd permutation. Note that we have included explicitly in the above an extra term, which takes explicit care of the identity operator in the spectrum, with $C_{ii\mathbb{1}} = 1$. The averaged four-point correlation function is crossing symmetric under the averaging,

$$\sum_m \overline{C_{ijm}C_{klm}} \left| \begin{array}{c} i \\ \\ j \end{array} \diagdown\!\!\!\!\diagup m \diagup\!\!\!\!\diagdown \begin{array}{c} l \\ \\ k \end{array} \right|^2 - \sum_n \overline{C_{lin}C_{jkn}} \left| \begin{array}{c} i \quad l \\ \diagdown\!\!\!\diagup \\ n \\ \diagup\!\!\!\diagdown \\ j \quad k \end{array} \right|^2 = 0. \tag{2.22}$$

However, higher moments of the averaged four-point functions still violate crossing symmetry if the distribution is simply Gaussian [12, 17]. For example, the average of the square of the crossing equation is non zero,

$$\overline{\left(G^s_{1221}(x, \bar{x}) - G^t_{1221}(x, \bar{x})\right)^2} \neq 0, \tag{2.23}$$

where $G^s_{ijkl}(x, \bar{x})$ and $G^t_{ijkl}(x, \bar{x})$ are defined as the $s$-channel expansion (A.6) and $t$-channel expansion (A.7) respectively. 1, 2 represents fixed external operators $\mathcal{O}_{1,2}$. If the crossing symmetry is retained perfectly, generally one would like to impose for any integer $n$,

$$\overline{\left(G^s_{ijkl}(x, \bar{x}) - G^t_{ijkl}(x, \bar{x})\right)^n} = 0. \tag{2.24}$$

For $n = 2$, equality is ensured if one introduces 4-th moment of the OPE statistics [12, 17]

$$\overline{C_{ijm}C_{mkl}C^*_{lin}C^*_{njk}}\Big|_c =$$
$$\left| \sqrt{C_0(P_i, P_j, P_m)C_0(P_m, P_k, P_l)C_0(P_l, P_i, P_n)C_0(P_n, P_j, P_k)} \left\{ \begin{array}{ccc} P_i & P_j & P_m \\ P_k & P_l & P_n \end{array} \right\} \right|^2. \tag{2.25}$$

Here the subscript $c$ represents the connected contribution on top of the Gaussian contribution,[5] with the Virasoro $6j$ symbol defined in (A.15). In [24], the authors explain

---

[5]To simplify the expression we do not write the most general expression which contains the delta function and signed permutations as in the Gaussian variance (2.20).

if they supplement the Gaussian ensemble with this 4-th moment, the ensemble average of $\langle G^s_{1234}(z, \bar{z}) G^t_{1234}(z, \bar{z}) \rangle$ will coincide with the computation by four-boundary wormhole and implement the crossing symmetry.

Furthermore, it appears that enforcing crossing symmetry for the ensemble-averaged four-point function does not necessarily guarantee crossing symmetry for ensemble-averaged higher-point functions [17, 18], in contrast to what holds in an exact CFT [25, 26]. We examine this point thoroughly in Appendix A.3 and explain how to restore the crossing symmetry at higher point correlation functions by introducing higher moments. For example, (2.25) is also required to restore the crossing symmetry in six-point correlation function.

### 2.3.2 Ensemble average of BCO structure coefficients

Since the structure coefficients exhibit similar universal formula (2.16), we can also introduce the ensemble of BCFT following the same spirit. Practically, we describe the ensemble average by specifying the (higher) variance of the random BCO structure coefficients. We assume that

$$\overline{C_{ijk}^{\alpha\beta\gamma}} = 0. \tag{2.26}$$

and propose that the variance should take the analogous form of (2.20) as,

$$
\begin{aligned}
\overline{C_{ijk}^{\alpha\beta\gamma} C_{nml}^{\sigma\rho\lambda}} = f_{ijk}^{\alpha\beta\gamma} f_{nml}^{\sigma\rho\lambda} + \\
C_0(P_i, P_j, P_k) \left( \delta_{\alpha\lambda}\delta_{\beta\rho}\delta_{\gamma\sigma}\delta_{il}\delta_{jm}\delta_{kn} + \delta_{\alpha\rho}\delta_{\beta\sigma}\delta_{\gamma\lambda}\delta_{in}\delta_{jl}\delta_{km} + \delta_{\alpha\sigma}\delta_{\beta\lambda}\delta_{\gamma\rho}\delta_{im}\delta_{jn}\delta_{kl} \right),
\end{aligned}
\tag{2.27}
$$

with

$$f_{ijk}^{\alpha\beta\gamma} := \sqrt{g_\beta g_\gamma} \delta_{\alpha\beta}\delta_{ij}\delta_{k\mathbb{1}} + \sqrt{g_\gamma g_\alpha} \delta_{\beta\gamma}\delta_{jk}\delta_{i\mathbb{1}} + \sqrt{g_\alpha g_\beta} \delta_{\gamma\alpha}\delta_{ki}\delta_{j\mathbb{1}}. \tag{2.28}$$

Note that setting $\overline{C_{ijk}^{\alpha\beta\gamma}} = 0$ (or $\overline{C_{ijk}} = 0$) may seem unnatural. However, it has been argued that such terms are subleading in the large-$c$ limit. To illustrate this, consider the limit $P_H \to \infty$ with fixed $P_0$, in which the diagonal heavy-heavy-light boundary structure coefficient $C_{0HH}^{\alpha\beta\beta}$ takes the form:

$$C_{0HH}^{\alpha\beta\beta} \sim \frac{1}{\sqrt{g_\alpha g_\beta}} \left( C_{\chi\mathbb{1}}^{\alpha} C_{\chi0}^{\beta} \right) \frac{\mathbb{S}_{P_H P_\chi}[P_0]}{\rho_0(P_H)}. \tag{2.29}$$

This formula is not universal since it depends on the bulk operator $\mathcal{O}_\chi$, the lightest bulk operator that couples to $\Psi_0^{\beta\beta}$. Nevertheless, if we take $P_H \to \infty$ the expression is indeed exponentially suppressed, supporting the above claim in [9]. Taking the above considerations into account, we propose (2.27). We assume $\overline{C_{ijk}^{\alpha\beta\gamma}} = 0$, while introducing an additional term $f_{ijk}^{\alpha\beta\gamma}$ in the variance to explicitly capture contributions from the identity operator. In fact, $f_{ii\mathbb{1}}^{\alpha\alpha\beta} = C_{ii\mathbb{1}}^{\alpha\alpha\beta}$. Similar to the closed case, this modification is crucial for preserving crossing symmetry, and we will demonstrate this explicitly in this context shortly below.

We comment that, in the ensemble-averaged results, the boundary conditions are completely factorized from the primary labels in the BCFT structure coefficients, unlike in

exact CFTs.[6] As a result, the variance we propose in (2.27) can be viewed as essentially a "half" or a "single" copy of that in the closed sector. This is natural, as a BCFT contains only a single copy of the Virasoro algebra. We also note that spin is not a good quantum number for BCOs, and thus relations like (2.21) do not apply to them. The variance (2.27) is designed to preserve the cyclic symmetry (2.9) of the structure coefficients.

The variance (2.27) also respects crossing symmetry of averaged BCFT four-point functions, which means for given external operators $i, j, k, l$ and boundary condition $\alpha, \beta, \gamma, \rho$, the averaged four point function satisfies

$$ \tag{2.30} $$

where the black line denotes the conformal block with the implicit summation in the intermediate channel. More explicitly,

$$\sum_m \frac{1}{\sqrt{g_\alpha g_\beta}} \overline{C_{ijm}^{\alpha\beta\gamma} C_{mkl}^{\rho\beta\alpha}} \quad \text{(diagram)} = \sum_n \frac{1}{\sqrt{g_\gamma g_\rho}} \overline{C_{inl}^{\rho\beta\gamma} C_{knj}^{\gamma\alpha\rho}} \quad \text{(diagram)} , \tag{2.31}$$

where the prefactors involving $g$-factors arise from the normalization of two-point functions, and we have canceled the common normalization factor $1/(g_\alpha g_\beta g_\gamma g_\rho)^{1/2}$ for the BCOs on both sides of the equation. The non-trivial case that needs to be verified is when $\alpha = \beta$, $\Psi_i^{\alpha\gamma} = \Psi_j^{\alpha\gamma} = \Psi_1^{\alpha\gamma}$, $\Psi_k^{\alpha\rho} = \Psi_l^{\alpha\rho} = \Psi_2^{\alpha\rho}$. Assigning the normalization correctly, we apply

---

[6]For example, in rational CFTs, they combine into the quantum $6j$ symbols [27, 28].

the variance formula (2.27) for the right hand side and it becomes

$$\sum_n \frac{1}{\sqrt{g_\gamma g_\rho}} \overline{C_{1n2}^{\rho\alpha\gamma} C_{2n1}^{\gamma\alpha\rho}} \;\; \raisebox{-1.5em}{[diagram: cross with legs 1,2 top and 1,2 bottom, internal $n$]} \;\; = \sqrt{g_\gamma g_\rho} \int_0^\infty dP_n \rho_0(P_n) C_0(P_1, P_2, P_n) \;\; \raisebox{-1.5em}{[diagram: cross with legs 1,2 top and 1,2 bottom, internal $n$]}$$

(2.32)

$$= \sqrt{g_\gamma g_\rho} \int dP_n \mathbb{F}_{\mathbb{1}P_n} \begin{bmatrix} P_1 & P_2 \\ P_1 & P_2 \end{bmatrix} \;\; \raisebox{-1.5em}{[diagram: cross with legs 1,2 top and 1,2 bottom, internal $n$]} = \sqrt{g_\gamma g_\rho} \;\; \raisebox{-1em}{[diagram: legs 1,2 left and 1,2 right joined by dashed $\mathbb{1}$ line]}\;.$$

As mentioned in the above review of large-$c$ asymptotics, we approximate the summation by the integral with the density of states $\rho_0(P_n)$, i.e.,

$$\sum_n \to \int dP_n g_\gamma g_\rho \rho_0(P_n).$$

(2.33)

This will be used repeatedly in the following. The left hand side of the equation becomes

$$\sum_n \frac{1}{g_\alpha} \overline{C_{11n}^{\alpha\alpha\gamma} C_{n22}^{\rho\alpha\alpha}} \;\; \raisebox{-1em}{[diagram: legs 1,2 with internal line $n$ horizontal]} = \sum_n \frac{1}{g_\alpha} \sqrt{g_\gamma g_\alpha} \sqrt{g_\alpha g_\rho} \delta_{n\mathbb{1}} \;\; \raisebox{-1em}{[diagram: legs 1,2 with internal line $n$ horizontal]}$$

(2.34)

$$= \sqrt{g_\gamma g_\rho} \;\; \raisebox{-1em}{[diagram: legs 1,2 left and 1,2 right joined by dashed $\mathbb{1}$ line]}\;,$$

which is exactly consistent with the right. Since the boundary indices factorise, this computation is essentially identical to checking crossing invariance of the four-point function of the closed sector for only a chiral half. As discussed in detail in section A.3, crossing in higher point correlation functions is violated unless we introduce non-Gaussianities. The discussion in section A.3 can be inherited directly here. The extra ingredient is to add appropriate boundary labels (also the correct normalization) and remove the square since we only have one copy of Virasoro algebra of the results in section A.3. Let us discuss the 4-th moment for $C_{ijk}^{\alpha\beta\gamma}$ as an example. Consider the 6 point correlation function with fixed boundaries and boundary changing operators, we can expand it into two distinct channels,

representing in the following graph

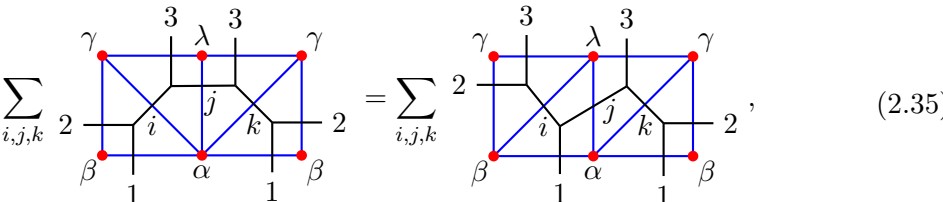

$$\sum_{i,j,k} \ \ = \ \sum_{i,j,k} \ \ , \qquad (2.35)$$

One can verify that if we require the equality exists up to level of the average value, in addition to Gaussian variance, we need to introduce the 4-th moment for the right hand side to balance the equation, i.e.,

$$\overline{C_{3jk}^{\alpha\gamma\lambda}C_{k12}^{\beta\gamma\alpha}C_{2i3}^{\lambda\gamma\beta}C_{1ji}^{\lambda\beta\alpha}}\Big|_c = \frac{1}{\sqrt{g_\alpha g_\beta g_\gamma g_\lambda}}\sqrt{\mathbf{C}_{3jk}\mathbf{C}_{k12}\mathbf{C}_{2i3}\mathbf{C}_{1ji}} \begin{Bmatrix} P_1 & P_2 & P_k \\ P_3 & P_j & P_i \end{Bmatrix}. \qquad (2.36)$$

where we use the abbreviation

$$\mathbf{C}_{ijk} := C_0(P_i, P_j, P_k). \qquad (2.37)$$

Following similar procedure described in Appendix A.3 we can fix the crossing symmetry violation problem level by level in the BCFT case.

### 2.3.3 Ensemble average of Open-Closed coupling coefficient

We also extend the discussion of ensemble average to include the open-closed coupling coefficients. Using the asymptotic formula (2.18) for bulk-to-boundary structure coefficient, we propose that its variance formula takes an analogous form[7]

$$\overline{C_{ij}^{\alpha}C_{kl}^{\beta}} = C_0(P_i, \bar{P}_i, P_j)\delta_{\alpha\beta}\delta_{ik}\delta_{jl}. \qquad (2.38)$$

Let us now verify that this ensures the ensemble-averaged BCFT two-point function on a cylinder is independent of the choice of channel decomposition. In a general BCFT, consider a cylinder with boundary conditions $\alpha$ and $\beta$, and insert two boundary operators $\Psi_i^{\alpha\alpha}$ and $\Psi_j^{\beta\beta}$ on the respective boundaries. There are two equivalent ways to decompose

---

[7]This perspective on random open-closed coupling coefficients has also been discussed in [29].

the corresponding path integral, analogous to (A.35) and (A.36)

$$\langle \Psi_i^{\alpha\alpha} \Psi_j^{\beta\beta} \rangle_{\text{cyl}} = \sum_{\Psi_k^{\alpha\beta}, \Psi_l^{\alpha\beta} \in \mathcal{H}_{\text{open}}^{\alpha\beta}} \frac{1}{g_\alpha g_\beta} C_{jlk}^{\alpha\beta\beta} C_{lki}^{\alpha\alpha\beta} \left( j \; \alpha \bigcirc i \right) \beta$$

$$= \sum_{\mathcal{O}_m \in \mathcal{H}_{\text{closed}}} C_{mi}^{\alpha} C_{mj}^{\beta} \left( j \; \alpha \bigcirc i \right) \beta \; .$$

(2.39)

Now that these OPE coefficients are random variables, we need to check whether the constraint remains satisfied after averaging. Given the variance (2.27) and (2.38), the non-trivial case is when $\alpha = \beta$, $i = j \neq \mathbb{1}$. Following the notation in [16] we define the conformal block in the open channel as $\mathcal{F}^{(\partial\text{bagel})}(P_k, P_l; P_i)$ and the conformal block in the closed channel as $\mathcal{F}^{(\partial\text{necklace})}(P_m, \bar{P}_m; P_i)$, and we suppress the moduli dependence (the length of the cylinder and the relative position to insert operators) explicitly. The first line will give

$$\sum_{k,l} \frac{1}{g_\alpha^2} \overline{C_{ilk}^{\alpha\alpha\alpha} C_{ilk}^{\alpha\alpha\alpha}} \mathcal{F}^{(\partial\text{bagel})}(P_k, P_l; P_i) = \sum_{k,l} \delta_{k\mathbb{1}} \delta_{li} \mathcal{F}^{(\partial\text{bagel})}(P_k, P_l; P_i) = \mathcal{F}^{(\partial\text{bagel})}(\mathbb{1}, P_i; P_i),$$

(2.40)

while the second line will give

$$\sum_m \overline{C_{mi}^\alpha C_{mi}^\alpha} \mathcal{F}^{(\partial\text{necklace})}(P_m, \bar{P}_m; P_i)$$
$$= \int_0^\infty dP_m d\bar{P}_m \rho_0(P) \rho_0(\bar{P}_m) C_0(P_m, \bar{P}_m, P_i) \mathcal{F}^{(\partial\text{necklace})}(P_m, \bar{P}_m; P_i) .$$

(2.41)

The kernel relates these two conformal blocks are written in the form

$$\mathcal{F}^{(\partial\text{bagel})}(P_k, P_l; P_i) = \int_0^\infty dP_m d\bar{P}_m \mathbb{K}_{P_m \bar{P}_m; P_k P_l}^{(\text{cyl-2pt})}[P_i] \mathcal{F}^{(\partial\text{necklace})}(P_m, \bar{P}_m; P_i) .$$

(2.42)

The exact form of the kernel $\mathbb{K}_{P_m \bar{P}_m; P_k P_l}^{(\text{cyl-2pt})}[P_i]$ is complicated. Fortunately, for $P_k = \mathbb{1}$ and $P_l = P_i$ we have

$$\mathbb{K}_{P_m \bar{P}_m; \mathbb{1} P_i}^{(\text{cyl-2pt})}[P_i] = \rho_0(P_m) \rho_0(\bar{P}_m) C_0(P_m, \bar{P}_m, P_i).$$

(2.43)

Therefore, equation (2.41) reduces to the result $\mathcal{F}^{(\partial\text{bagel})}(\mathbb{1}, P_i; P_i)$, which satisfies the bootstrap constraint as desired.

## 3 Ensemble average of multi-copy BCO correlation functions

In this section, we consider ensemble average of multi-copy BCO correlation functions, and show the connection to the Virasoro TQFT.

In [9] the authors compute the ensemble average of multi-copy correlation functions for CFT bulk operators. It is observed that the results in the large-$c$ limit agrees with the on-shell pure gravitational action computation of associated Euclidean wormhole [10]. In [14, 24] the authors extended the result to quantum level using the Virasoro TQFT and get more interesting results. In particular, they identify non Gaussianity of the ensemble average with multi-boundary wormhole in gravity. These considerations can be readily generalised to averages of BCO correlation functions. In this part we will present several examples, and comment on their relations between Virasoro TQFT and wormholes. Related discussions appear also in [21, 30].

Consider two copies of the same disk, each being a BCO three point correlation function. They share the same conformal boundary conditions and BCO insertions.

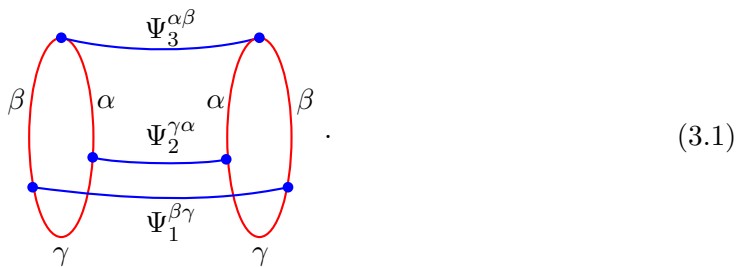

$$(3.1)$$

The above correlation functions give

$$
\frac{1}{g_\alpha g_\beta g_\gamma} \overline{\langle \Psi_1^{\beta\gamma}(\infty)\Psi_2^{\gamma\alpha}(1)\Psi_3^{\alpha\beta}(0)\rangle \langle \Psi_3^{\beta\alpha}(\infty)\Psi_2^{\alpha\gamma}(1)\Psi_1^{\gamma\beta}(0)\rangle}
$$
$$
= \frac{1}{g_\alpha g_\beta g_\gamma} \overline{C_{123}^{\alpha\beta\gamma} C_{321}^{\gamma\beta\alpha}} = \frac{1}{g_\alpha g_\beta g_\gamma} C_0(P_1, P_2, P_3).
$$

$$(3.2)$$

The $g$-factors arise from the normalization of correlation functions using two-point functions. Up to normalization, this coincides with the partition function of the Virasoro TQFT on a three-punctured sphere times an interval [24].

$$
Z_{\text{Vir}}\left( \vcenter{\hbox{\includegraphics{spheres}}} \right) = C_0(P_1, P_2, P_3).
$$

$$(3.3)$$

From the perspective of AdS/CFT correspondence this is unsurprising. The gravitational path integral $Z_{\text{grav.}} \sim |Z_{\text{Vir}}(M)|^2$ and it is equal to the partition $Z_{\text{CFT}}$ of the CFT living on the boundary of the bulk manifold. BCO correlation functions generally take the form of a chiral half of the closed CFT. Hence the result from ensemble average of BCFT should be the "square root" of the closed CFT, which matches directly with one copy of the Virasoro TQFT computation.

In a more interesting example the four-boundary wormhole emerges. Consider the following configuration of disk three point BCO correlation functions with various edges and conformal boundaries shared between disks.

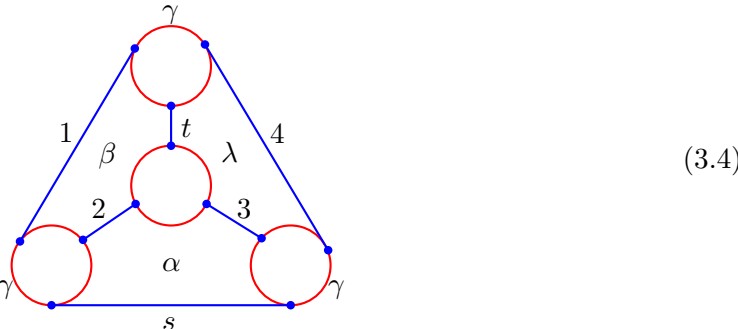

$$(3.4)$$

The ensemble average of the above follows directly from the four-moment (2.36)

$$\frac{1}{(g_\alpha g_\beta g_\gamma g_\lambda)^2} \sqrt{\mathbf{C}_{12s}\mathbf{C}_{34s}\mathbf{C}_{14t}\mathbf{C}_{23t}} \begin{Bmatrix} P_1 & P_2 & P_s \\ P_3 & P_4 & P_t \end{Bmatrix},$$

$$(3.5)$$

which exactly reproduces the Virasoro TQFT result for four-boundary non-Gaussianity wormhole up to the previous normalization factor

$$Z_{\text{Vir}}\left( \vcenter{\hbox{\includegraphics{wormhole}}} \right) = \sqrt{\mathbf{C}_{12s}\mathbf{C}_{34s}\mathbf{C}_{14t}\mathbf{C}_{23t}} \begin{Bmatrix} P_1 & P_2 & P_s \\ P_3 & P_4 & P_t \end{Bmatrix}.$$

$$(3.6)$$

This suggests that we could use one copy of Virasoro TQFT to compute the higher moments of structure coefficients $C_{ijk}^{\alpha\beta\gamma}$, which is expected to produce solutions to the bootstrap constraints of higher-point correlation functions. To read off the correspondence between the results of ensemble average with 3d geometries, for every disk three point function in the ensemble average one replaces it with the three punctured sphere

$$C_{ijk}^{\alpha\beta\gamma} = \vcenter{\hbox{\includegraphics{disk}}} \longrightarrow \vcenter{\hbox{\includegraphics{sphere}}}$$

$$(3.7)$$

and draw the corresponding Euclidean multi-boundary wormhole $\mathcal{M}$. The correct answer for the higher moment will be upto normalisation given by the normalisation of the BCOs and also factors of $g_\alpha$ from each conformal boundary condition, the $Z_{\text{Vir}}(\mathcal{M})$. Another

remark is that, since BCOs do not have braiding structure, certain Virasoro TQFT results involving non-trivial braiding vanish in BCFT. This is encoded in the averaged two point correlation function of BCO structure coefficients in (2.27). OPE coefficients with exchanged primary indices (as opposed to cyclic permutation of the primary along with boundary indices) are considered as independent variables, rather than related by phases following from their braiding as in (2.21). Graphically speaking, one does not allow the blue lines, representing the BCOs insertion to intersect.

We can also consider higher point disk correlation functions. This introduces moduli into the manifold. Consider a four point BCO correlation function $\langle \Psi_1^{\alpha\beta} \Psi_2^{\beta\gamma} \Psi_2^{\gamma\beta} \Psi_1^{\gamma\alpha} \rangle$, with fixed operator $\Psi_3^{\alpha\gamma}$ in the intermediate channel

$$= \frac{1}{g_\beta \sqrt{g_\alpha g_\gamma}} C_0(P_1, P_2, P_3) \;\; \begin{smallmatrix}1\\ \\2\end{smallmatrix} \!\!\diagdown\!\!\!\!\diagup\!\! 3 \!\!\diagdown\!\!\!\!\diagup\!\! \begin{smallmatrix}1\\ \\2\end{smallmatrix} \;.$$

This is quite similar to the handle wormhole result derived from Virasoro TQFT,

$$Z_{\mathrm{Vir}} \left( \;\cdots\; \right) = C_0(P_1, P_2, P_3) \;\; \begin{smallmatrix}1\\ \\2\end{smallmatrix} \!\!\diagdown\!\!\!\!\diagup\!\! 3 \!\!\diagdown\!\!\!\!\diagup\!\! \begin{smallmatrix}1\\ \\2\end{smallmatrix} \;, \tag{3.9}$$

whose outer boundary is a four punctured sphere and inner boundary consists of two identified three-punctured sphere. Note however, in BCFT correlation functions, the moduli that would appear in the conformal blocks are real. For example, the moduli in (3.8) appearing in the conformal block is the real cross ratio $\eta$ in (A.24), whereas the closed conformal blocks depend on a generically complex moduli.

One can easily generalize to other examples. For instance, consider $k$ copies of 4 point correlation function with $2k$ BCO insertions

$$Z_k := G_k^{-1} \times$$
$$\overline{\langle \Psi_1^{\alpha_1\alpha_2} \Psi_2^{\alpha_2\alpha_3} \Psi_3^{\alpha_3\alpha_4} \Psi_4^{\alpha_4\alpha_1} \rangle \langle \Psi_4^{\alpha_1\alpha_4} \Psi_3^{\alpha_4\alpha_3} \Psi_5^{\alpha_3\alpha_5} \Psi_6^{\alpha_5\alpha_1} \rangle \cdots \langle \Psi_{2k}^{\alpha_1\alpha_{2k-1}} \Psi_{2k-1}^{\alpha_{2k-1}\alpha_3} \Psi_2^{\alpha_3\alpha_2} \Psi_1^{\alpha_2\alpha_1} \rangle},$$
$$\tag{3.10}$$

where $G_k$ is the normalization factor

$$G_k := (g_{\alpha_1} g_{\alpha_2} g_{\alpha_3} g_{\alpha_4})^{\frac{1}{2}} (g_{\alpha_1} g_{\alpha_4} g_{\alpha_3} g_{\alpha_5})^{\frac{1}{2}} \cdots (g_{\alpha_{2k-1}} g_{\alpha_3} g_{\alpha_2} g_{\alpha_1})^{\frac{1}{2}} \;. \tag{3.11}$$

The graphical representation (take $k = 3$ as an example) is given by

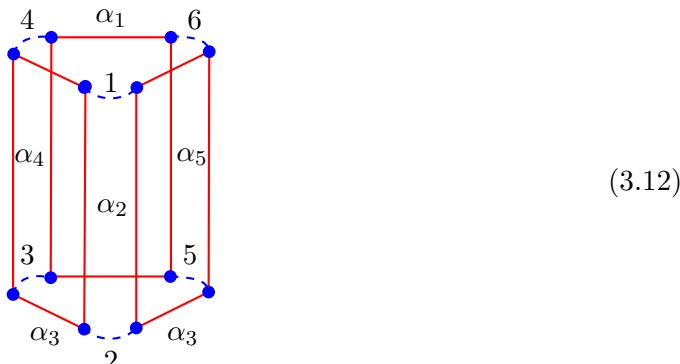

$$(3.12)$$

Expanding the correlation function in the same channel we will have

$$Z_{k=3} = \frac{G_3^{-1}}{(g_{\alpha_1} g_{\alpha_3})^{3/2}} \sum_{p_1, p_2, p_3} \overline{C_{12p_1}^{\alpha_3 \alpha_1 \alpha_2} C_{p_3 21}^{\alpha_2 \alpha_1 \alpha_3}} \; \overline{C_{34p_1}^{\alpha_1 \alpha_3 \alpha_4} C_{p_2 43}^{\alpha_4 \alpha_3 \alpha_1}} \; \overline{C_{56p_3}^{\alpha_3 \alpha_1 \alpha_5} C_{p_2 65}^{\alpha_5 \alpha_1 \alpha_3}} \times$$

$$= \frac{G_3^{-1}}{(g_{\alpha_1} g_{\alpha_3})^{1/2}} \int_0^\infty dP \rho_0(P) C_0(P_1, P_2, P) C_0(P_3, P_4, P) C_0(P_5, P_6, P)$$

$$(3.13)$$

which is again similar to the result of Euclidean cyclic wormhole computed by Virasoro TQFT.

## 4 Ensemble average of the closed CFT path-integral from triangulation

It was proposed that the path-integral of a 2D CFT can be expressed as a discrete tensor network made up of BCO correlation functions [1–3]. This is reviewed in Appendix B. Since the path integral is a product of boundary structure coefficients, it suggests that in the large-$c$ limit, the path-integral should exhibit universal behavior following from the average of the collection of BCO OPE coefficients defined in the previous section. In the following, we will consider the leading large-$c$ averages of different 2D CFT path-integrals.

### 4.1 Path-integral over a single connected manifold with no insertion

Recalling that in the discretisation of the path-integral reviewed in (B.8), we assign the vacuum Ishibashi state $|\mathbb{1}\rangle\rangle$ at the rim of each hole, which is expressed as a weighted sum of Cardy states $|B_\alpha\rangle$. This weighted sum has advantages related to considerations from generalized symmetries [31, 32], and it projects out relevant perturbations. In this paper, however, we will adopt a simpler treatment of the holes. Note that each Cardy state can

be written as

$$|B_\alpha\rangle = g_\alpha \, |1\rangle\rangle + \sum_{i \in s_{gap}} \mathcal{B}_\alpha^i \, |P_i\rangle\rangle \,, \tag{4.1}$$

where we assume that there is a gap $s_{\text{gap}}$ in the spectrum above the vacuum state. In the small hole size limit considered in this paper—analogous to the high-temperature limit in the derivation of the Cardy formula [22]—the contributions from these operators are suppressed, and we can approximately express the Ishibashi state as

$$|1\rangle\rangle \sim g_\alpha^{-1} \, |B_\alpha\rangle \,. \tag{4.2}$$

Similar to the extension of validity for the density of states in the large-$c$ limit [23], we expect the suppression of non-vacuum Ishibashi states to be further enhanced at large $c$. Consequently, each Cardy state approximates the vacuum Ishibashi state up to normalization with even better accuracy in the large-$c$ limit, and we will not use the weighted sum over Cardy states to recover the vacuum Ishibashi state in this paper. The path integral expression then reduces to

$$Z_{\mathcal{M}} = \lim_{R \to 0} \mathcal{N}(R) \left( \prod_{\alpha \in \{\alpha_v\}} g_\alpha \right)^{-1} Z_{\{\alpha_v\}}, \quad Z_{\{\alpha_v\}} = \sum_{\{i,I\}} \prod_\Delta Z^{\alpha\beta\gamma}_{(k,K),(i,I),(j,J)} \,. \tag{4.3}$$

where $R$ is the size of the holes and $\mathcal{N}(R)$ is a normalization factor as specified in (B.15). As we discussed in the appendix, these normalization factors arise from shrinking the holes. They have no impact on our discussion and are often omitted in what follows. Now all the OPE coefficients are random variables, and we can compute the ensemble average of this product of OPEs. More precisely, we should compute

$$\overline{Z_{\mathcal{M}}} = \lim_{R \to 0} \mathcal{N}(R) \left( \prod_{\alpha \in \{\alpha_v\}} g_\alpha \right)^{-1} \sum_{\{i,I\}} \prod_\Delta \overline{C^{\alpha\beta\gamma}_{ijk} \frac{\gamma^{IJK}_{ijk}}{\sqrt{g^{\alpha\beta}_k g^{\beta\gamma}_i g^{\gamma\alpha}_j}}} \,, \tag{4.4}$$

where we have used the assumption that $g_\alpha$ are independent variables from the BCO structure coefficients, and thus can be moved out of the averaging freely.

Contrasting the computations in section 2, where the external operators are fixed, in the path integral (4.4) we need to sum over all the degrees of freedom and we need to be careful about what principles should be applied during the computation. Recalling the variance (2.27) or the higher moment (2.36), where the factor $g_\alpha$ only enters as an overall normalization, it is expected that the dependence on the boundary labels will be completely factorized after taking the average.

To manipulate the rest of the expression, we need to pick a specific triangulation to start with. A natural question is whether the result is independent of the triangulation. For rational CFTs or the Liouville CFT, given the boundary $\alpha, \beta, \gamma, \rho$ and external primary

operators $i, j, k, l$, one can show that [1–3]

$$Z_{ijkl} := \sum_\lambda w_\lambda$$ 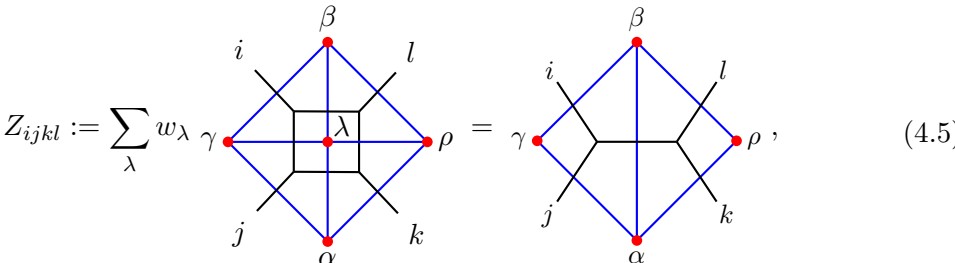

which can be proved via crossing symmetry, the Pentagon equations and orthogonal relation encoded in the $6j$ symbols. It was shown that the above equality is sufficient to convert any triangulation to any other. However, the ensemble average allows fluctuations violating crossing symmetry in a mild way. Independence based on crossing symmetry is thus no longer taken for granted, and should be explored with care. Let us examine (4.5) in our setting and illustrate that the independence of triangulation still holds. Once again, it is only meaningful that the equation holds in the sense of averaging, i.e.,

$$\frac{g_\lambda^{-1}}{g_\lambda^2 \sqrt{g_\alpha g_\beta g_\gamma g_\rho}} \sum_{m,n,r,s} \overline{C_{msi}^{\beta\gamma\lambda} C_{jnm}^{\lambda\gamma\alpha} C_{rnk}^{\alpha\rho\lambda} C_{lsr}^{\lambda\rho\beta}} \quad$$

$$= \frac{1}{\sqrt{g_\alpha g_\beta}} \sum_m \overline{C_{ijm}^{\alpha\beta\gamma} C_{mkl}^{\rho\beta\alpha}} \qquad . \tag{4.6}$$

Let us first explain the normalization factor before the expression in the left hand side of the equation. To avoid mess we have simultaneously canceled the normalization factor $(g_\alpha g_\beta g_\gamma g_\rho)^{-1/2}$ of the external operators $i, j, k, l$ on both sides of the equations. The remaining normalization factor in the left hand side comes from intermediate operator insertions, which give $g_\lambda^{-2} (g_\alpha g_\beta g_\gamma g_\rho)^{-1/2}$. The extra $g_\lambda^{-1}$ factor comes from expressing the Cardy boundary state in terms of the vacuum Ishibashi state as in (4.3). Previously we have computed the right hand side in equation (2.34) which does not vanish when $\alpha = \beta$, $i = j = 1$ and $k = l = 2$[8]. Hence, what we need to check is

$$\frac{1}{g_\lambda^3 g_\alpha \sqrt{g_\gamma g_\rho}} \sum_{m,n,r,s} \overline{C_{ms1}^{\alpha\gamma\lambda} C_{1nm}^{\lambda\gamma\alpha} C_{rn2}^{\alpha\rho\lambda} C_{2sr}^{\lambda\rho\beta}} \qquad . \tag{4.7}$$

[8]Again, the labels $1, 2$ corresponds to operators $\Psi_1^{\alpha\gamma}$, and $\Psi_2^{\alpha\rho}$ respectively.

Noting the arrangement of these indices, by using our variance formula (2.27) and Wick contractions, we can simplify further

$$
\frac{1}{g_\lambda^3 g_\alpha \sqrt{g_\gamma g_\rho}} \sum_{m,n,r,s} \overline{C_{ms1}^{\alpha\gamma\lambda} C_{1nm}^{\lambda\gamma\alpha}} \, \overline{C_{rn2}^{\alpha\rho\lambda} C_{2sr}^{\lambda\rho\beta}} \quad \text{}
$$

$$
= \frac{1}{g_\lambda^3 g_\alpha \sqrt{g_\gamma g_\rho}} \sum_{m,n,r,s} \mathbf{C}_{ms1} \mathbf{C}_{rn2} \delta_{sn} \quad \text{}
$$

(4.8)

$$
= \sqrt{g_\gamma g_\rho} \int_0^\infty dP_m dP_n dP_r \, \rho_0(P_m) \rho_0(P_n) \rho_0(P_r) \mathbf{C}_{ms1} \mathbf{C}_{rn2} \quad \text{}
$$

$$
= \sqrt{g_\gamma g_\rho} \int_0^\infty dP_n \, \rho_0(P_n) \quad \text{} \; ,
$$

where in the third line we replace the discrete summation to the continuous integral over states above threshold. The crossing kernel (A.12) is used to obtain the last equality. Repeating similar computation in [3], we isolate the loop from the rest by crossing relations

$$
\sqrt{g_\gamma g_\rho} \int_0^\infty dP_n \, \rho_0(P_n) \quad \text{} \; = \sqrt{g_\gamma g_\rho} \int_0^\infty dP_n \, \rho_0(P_n) \quad \text{}
$$

(4.9)

$$
= \sqrt{g_\gamma g_\rho} \quad \text{} \; ,
$$

which completely agrees with (2.34). Here we take the small hole limit in the last step and ignore the regularization factor coming from this hole shrinking operation, as described in (B.13). Thus, we have demonstrated that in the context of ensemble averages—given the external operators and boundary conditions—the triangulation made up of four triangles and the triangulation composed of two triangles (4.5) yield the same result.

Nevertheless, this is not sufficient for complete triangulation independence. It is necessary to verify that the manipulations above (i.e. equation (4.5)) hold locally when the above graph forms a subregion of a higher point correlation function to be averaged. As discussed in detail in Section 2 and Appendix A.3, local crossing symmetry embedded within higher-point correlation functions is not automatically guaranteed by enforcing crossing symmetry only at the level of four-point functions. In the following we carefully examine the triangulation independence by considering diagrams involving more triangles.

**Leading-order diagram and loop model**

Consider discretisation of the path-integral with a regular triangulation as shown in Figure 1a). Such triangulations naturally contain building blocks involving $\overline{Z_{ijkl}}$ discussed in the previous section. Therefore without loss of generality we can consider manipulating this patch, assuming that it is part of a larger graph. We will return to triangulation independence of the path-integral shortly. Let us use $\mathfrak{T}$ to denote the sets of all triangles (see Figure 1a). From what we have discussed, we know that the path integral can be written

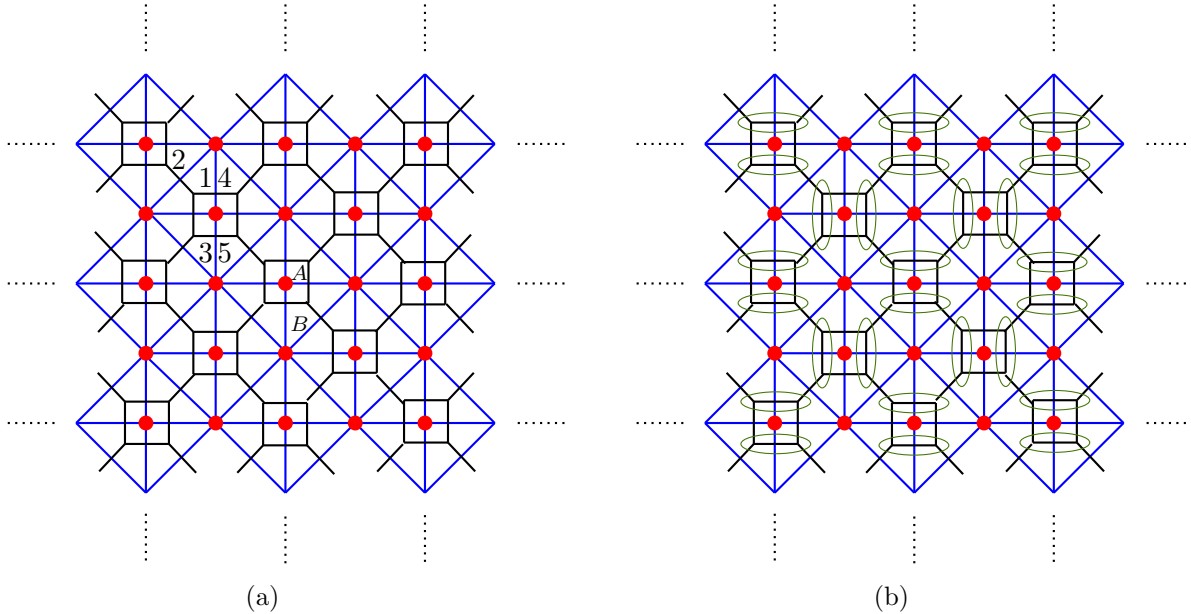

(a)            (b)

**Figure 1**. (a) The denser triangulations. One can think that the diagram we draw represents the portion far away from the edges of the triangulations. This triangulation contains two types of vertices, $A$ and $B$. Type $A$ vertices are shared by four triangles while Type $B$ vertices are shared by eight. (b) A possible two-two pairing $\mathcal{P}$ where two triangles form a pair if they are circled in green. In particular, this diagram is actually an example of the leading order diagram.

in the form of (4.4)

$$\overline{Z_{\mathcal{M}}} = \left(\prod_{\alpha \in \alpha_v} g_\alpha\right)^{-1} \sum_{\{a_i, A_i\}} \overline{\prod_{i \in \mathfrak{T}} C^{\alpha_i \beta_i \gamma_i}_{a_i b_i c_i}} \frac{\gamma^{A_i B_i C_i}_{a_i b_i c_i}}{\sqrt{g_{\alpha_i} g_{\beta_i} g_{\gamma_i}}}, \qquad (4.10)$$

where we use label $i$ to represent the $i$-th triangle in the set $\mathfrak{T}$ and $\alpha_i, \beta_i, \gamma_i, a_i, b_i, c_i$ are the corresponding edge and corner labels of the $i$-th triangle. The key point is to simplify the average value of the product of open structure coefficients. Given the specific triangulation shown in figure 1a, by Wick contraction we have

$$\overline{\prod_{i \in \mathfrak{T}} C_{a_i b_i c_i}^{\alpha_i \beta_i \gamma_i}} = \sum_{\mathcal{P} \in \mathfrak{P}} \prod_{(i_1, i_2) \in \mathcal{P}} \overline{C_{a_{i_1} b_{i_1} c_{i_1}}^{\alpha_{i_1} \beta_{i_1} \gamma_{i_1}} C_{a_{i_2} b_{i_2} c_{i_2}}^{\alpha_{i_2} \beta_{i_2} \gamma_{i_2}}}. \tag{4.11}$$

Let us set the total number of triangles to be $\mathcal{N}$, with $\mathcal{N}$ even. Here $\mathcal{P}$ is one of the configurations among $(\mathcal{N}-1)!!$ possible two-two pairings. We use $\mathfrak{P}$ to denote the sets of these configurations. For example, suppose the triangles are labeled by $\{1, 2, \ldots, \mathcal{N}-1, \mathcal{N}\}$, one possible pairing $\mathcal{P}$ in $\mathfrak{P}$ is $\{(1,2), (3,4), \ldots, (\mathcal{N}-1, \mathcal{N})\}$. Figure 1b depicts one of the possible pairing configurations.

In the path integral with BCO insertions, all the indices representing primaries and descendants need to be summed. Notice the structure of the variance (2.27), which includes many Kronecker delta functions, and each term is essentially constraining the labels on the edges, or equivalently, restrictions on the phase space over which one performs integration. As a result, the term that leads to the fewest restrictions on the integration phase space would be enhanced by a larger phase space volume. Such phase space enhancements are maximum when there is the largest number of constraints automatically satisfied in the configuration following from the discretisation. For example, neighbouring triangles share at least one edge, whose primary labels match by construction. Thus constraints from the ensemble average imposing their equality do not lead to a reduction of the integration phase space, giving the leading contributions.

Meanwhile, there is a very important exception to the above counting, though a bit counterintuitive. Namely, one should not view the delta function containing the identity, such as $\delta_{a\mathbb{1}}$, as an extra restriction on the phase space. We can clearly see this from the previous calculation (Eq. (2.32) and (2.34)). In (2.32), all the delta functions are trivially satisfied, and after summing over the intermediate primary operators, the result is the same as in Equation (2.34), where in the dual channel the operator is set to the identity. In other words, crossing symmetry suggests that imposing an intermediate edge to the identity should contribute at the same order as integrating an edge through $P$ above threshold. Therefore, to single out the genuine constraints on phase space, we should only count the terms containing delta functions that do not involve the identity.

With this principle, we can further simplify the equation (4.11). It is obvious that adjacent triangles share one common edge. In other words, at leading order, we only need to consider pairings of adjacent triangles two by two, denoted by $\hat{\mathfrak{P}}$, i.e.,

$$\overline{\prod_{i \in \mathfrak{T}} C_{a_i b_i c_i}^{\alpha_i \beta_i \gamma_i}} \approx \sum_{\mathcal{P} \in \hat{\mathfrak{P}}} \prod_{(i_1, i_2) \in \mathcal{P}} \overline{C_{a_{i_1} b_{i_1} c_{i_1}}^{\alpha_{i_1} \beta_{i_1} \gamma_{i_1}} C_{a_{i_2} b_{i_2} c_{i_2}}^{\alpha_{i_2} \beta_{i_2} \gamma_{i_2}}}. \tag{4.12}$$

For example, in the Figure 1a, triangle 1 can in principle pair with any other triangle. However, due to the phase space consideration, triangle 1 can only pair with 2, 3, or 4;

pairing with 5 is relatively suppressed. Previously, we mentioned that correlations exist between different OPE coefficients—even when they are far apart. This threatens locality of the CFT path-integral. However, it is clear that locality is actually preserved to leading order. The computation is also tremendously simpliefied since we only need to focus on the correlations between OPEs represented by adjacent triangles. In the following, approximately equality $\approx$ implies equality to the leading order.

Now the basic building blocks of a generic diagram with our choice of triangulation are the following two kinds of pairing of adjacent triangles

$$
\text{} \qquad (4.13)
$$

Substitute the expression of each triangle and apply the averaging we have

$$
\text{} \qquad (4.14)
$$

The first term here comes from the initial term in the sum, and we have bundled the remaining terms into the term $R_{ijkl}^{\alpha\beta\gamma\rho}$, which can be shown negligible shortly. For clarity, we have drawn the boundaries and corners of the triangles. The identity label corresponds to exchanging the identity primary family, which includes all the descendants in the family.

Similar computation for the second block will produce

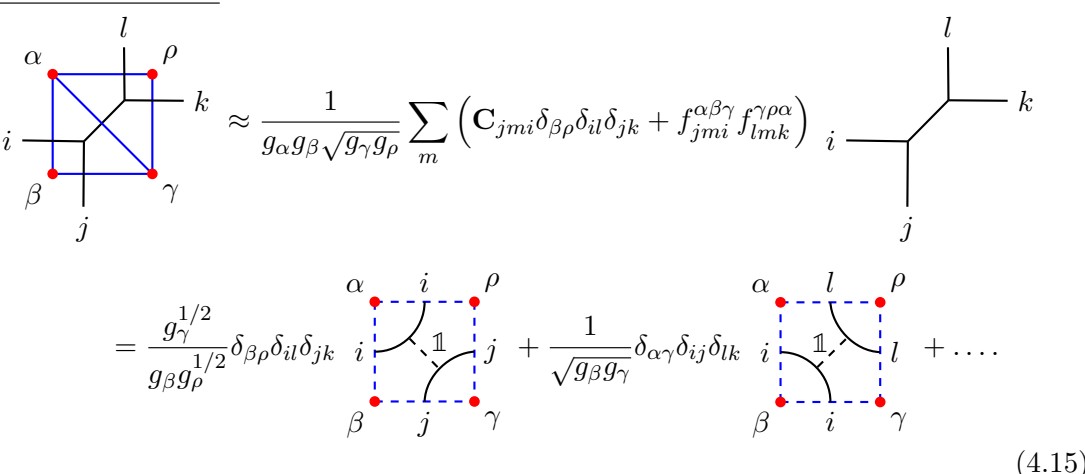

$$(4.15)$$

We will also explain what terms are in the ellipsis shortly.

Importantly, the ensemble average of our diagram reduces to one composed of these building blocks. The averaged result is reminiscent of loop models in integrable lattice models [19, 20], where the full transfer matrix is made up of squares composing of these parallel segments. The resultant diagram of conformal blocks (or the path integral) would then form a sum over closed loops since they have no where to end, unless there are external boundaries. Note that each closed loop corresponds to a trivial satisfaction of a constraint, which is analogous to analysis of matrix models in the large $N$ limit. The more closed loops there are, the more phase space enhancement there is. Hence, the leading-order diagram is the one that contains the maximum number of closed loops.

Now the problem has been converted to one of finding a configuration where adjacent triangles are paired in such a way that, after ensemble averaging, the maximum number of closed loops is obtained. We can think of this simple combinatorial problem as follows. In our grid, there are two types of vertices: Type $A$, which is shared by four triangles, and Type $B$, which is shared by eight triangles (see $A, B$ in figure 1a). It is evident that if we want to maximize the number of closed loops, these loops should be as short as possible. We define the length of a loop by the number of triangles it traverses. Based on this definition, a loop that encompasses only a Type $A$ vertex has a length of 4, while a loop that encompasses only a Type $B$ vertex has a length of 8. And incorporating more vertices necessitates longer loops. Therefore, one could single out the leading contribution by packing into the diagram all possible loops, each containing a single Type A vertex. The remaining loops are then filled, each containing one Type B vertex. It is evident that choosing the first term in the last line of (4.14) as the basic unit, we can use the following

square block to tile the entire triangulation

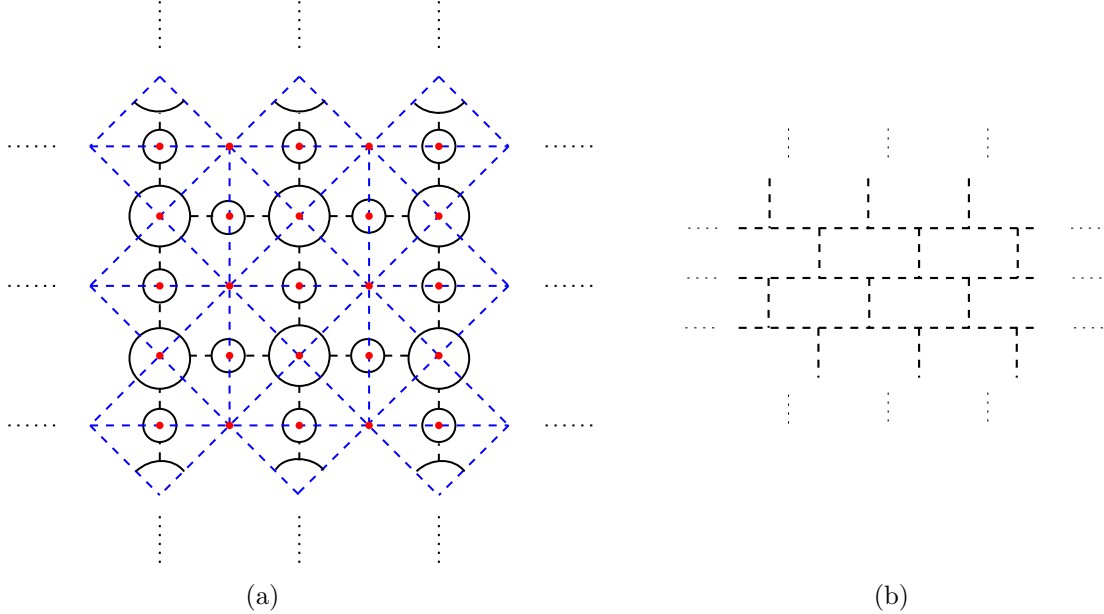

$$= \frac{\delta_{il}\delta_{jk}\delta_{\gamma\rho}}{\sqrt{g_\gamma g_\rho}} \int_0^\infty dP_m \rho_0(P_m) \quad m\bigcirc = \frac{\delta_{il}\delta_{jk}\delta_{\gamma\rho}}{\sqrt{g_\gamma g_\rho}} \qquad , \qquad (4.16)$$

which yields all the possible diagrams with the maximum number of closed loops. Notice that this is actually what we computed previously in (4.9), up to extra factors that follow from normalization factor of the $i, j, k, l$ external operators. Figure 1b is one of the possible configurations and it leads to the diagram shown on the left panel of the following figure.

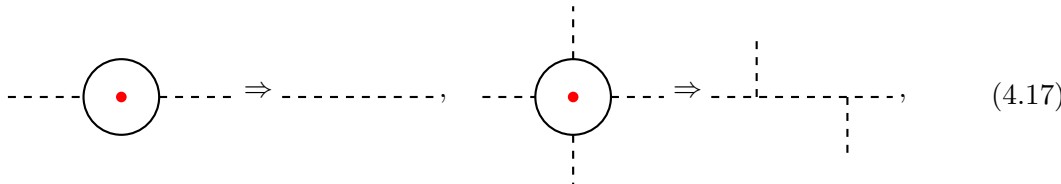

(a)                        (b)

**Figure 2**. (a) The leading-order diagram obtained by the pairing of Figure 1b, formed by the square block (4.16). (b) The identity network after contracting the closed loops.

Repeating the computation in (4.9), we can further simplify the diagram by shrinking the loops in the following way

$$\qquad \Rightarrow \qquad , \qquad \Rightarrow \qquad , \qquad (4.17)$$

which leads to the network of identity blocks shown in figure 2b.

**Contributions of $R_{ijkl}^{\alpha\beta\gamma\lambda}$**

We argued that a term denoted $R_{ijkl}^{\alpha\beta\gamma\lambda}$ in (4.15) is negligible to leading order. In the following we will justify this claim. Explicitly, substitute the equation (2.28) we get

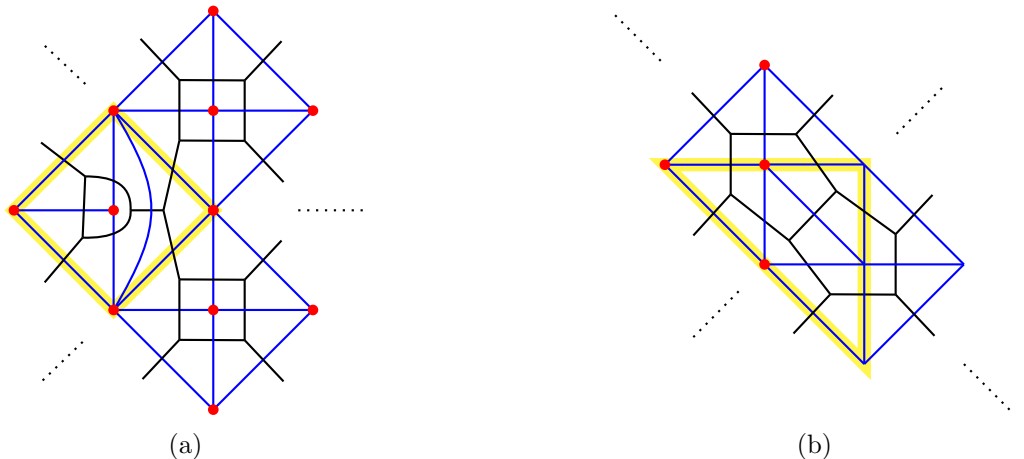

$$(4.18)$$

Similar computations can be done to obtain the terms in ellipses in (4.15). One can easily verify that if we use these building blocks ((4.15) and (4.18)), the resulting diagram contains fewer closed loops than the one shown in figure 2a. Therefore, in the large-$c$ limit, the contributions of diagrams composed of these blocks are relatively suppressed.

**Independence of triangulations**

In the following we would like to inspect any dependence of the ensemble averaged path-integral on triangulations. There are two aspects to be verified. First, whether the result is invariant for different triangulations with the same total number of triangles. More precisely, consider two triangulations related by crossing as shown in figure. (1a), In Ap-

**Figure 3**. Two possible configurations obtained by one crossing move of Figure 1a. (a) Obtained by crossing move of one of the building block (4.14). The highlight part produces a structure of 4-th moment (2.36). (b) Obtained by crossing move of one of the building block (4.15). Similarly, the highlight part produces 4-th moment. Introducing non-Gaussianity helps us retain the crossing symmetry.

pendix A.3 we explain how to maintain the crossing symmetry by adding higher moments.

Applying ensemble average of the configuration shown in figure 1a, these higher moments are not visible in the computation due to the fact that this triangulation stacks together conformal blocks in the same channel. However, when considering configurations shown in Figure 3, the higher moments are crucial for us. Let us explicitly evaluate the left configuration in Figure 3 as an illustration of this point. We can focus on the part highlighted in yellow, since other parts of the diagram remain in the pattern as in (4.16), i.e.,

$$
\text{[diagram]} = \frac{g_\beta^{-1}}{g_\alpha^{3/2} g_\beta^{3/2} g_\lambda^{3/2} g_\gamma g_\rho^{1/2}} \sum_{k,l,m,n} \overline{C_{lmn}^{\lambda\alpha\beta} C_{nji}^{\rho\alpha\lambda} C_{ikl}^{\beta\alpha\gamma} C_{jmk}^{\beta\gamma\lambda}} \quad \text{[diagram]}
$$

$$
= \frac{\delta_{\gamma\rho}}{\sqrt{g_\gamma g_\rho}} \int_0^\infty \left[ \prod_{a\in\{k,l,m,n\}} dP_a \rho_0(P_a) \right] \rho_0(P_k)^{-1} \mathbf{C}_{jin} \mathbf{C}_{nlm} \mathbb{F}_{P_n P_k} \begin{bmatrix} P_l & P_i \\ P_m & P_j \end{bmatrix} \quad \text{[diagram]} ,
$$

(4.19)

where we write the 4-th moment (2.36) in terms of the crossing kernel for convenience. The normalization reduces to be of order $g_\gamma^{-1}$. Notice that we can integrate over $P_k$ to get

$$
\frac{\delta_{\gamma\rho}}{\sqrt{g_\gamma g_\rho}} \int_0^\infty \left[ \prod_{a\in\{l,m,n\}} dP_a \rho_0(P_a) \right] dP_k \mathbf{C}_{jin} \mathbf{C}_{nlm} \mathbb{F}_{P_n P_k} \begin{bmatrix} P_l & P_i \\ P_m & P_j \end{bmatrix} \quad \text{[diagram]}
$$

$$
= \frac{\delta_{\gamma\rho}}{\sqrt{g_\gamma g_\rho}} \int_0^\infty \left[ \prod_{a\in\{l,m,n\}} dP_a \rho_0(P_a) \right] \mathbf{C}_{jin} \mathbf{C}_{nlm} \quad \text{[diagram]} .
$$

(4.20)

Since $\rho_0(l)\mathbf{C}_{nlm}$ is another crossing kernel (A.12), we can absorb it in a crossing transformation

$$
\frac{\delta_{\gamma\rho}}{\sqrt{g_\gamma g_\rho}} \int_0^\infty dP_n \rho_0(P_n) dP_m \rho_0(P_m) \mathbf{C}_{jin} \quad \text{[diagram]} = \frac{\delta_{\gamma\rho}}{\sqrt{g_\gamma g_\rho}} \quad \text{[diagram]} ,
$$

(4.21)

where we again apply the similar trick as we used in equation (4.9), assuming the hole is infinitesimal. We can see that it exactly matches with the result (4.16). Hence, we confirm that, in the presence of higher moments, different triangulations related by crossing

transformation produce the same results.

Second, we would like to check invariance when number of total triangles change. More specifically,consider a sparser grid compared to that in Figure 1a. The final result always yields a network composed solely of the identity. Since identities can fuse and undergo crossing move arbitrarily, it is not difficult to prove that the resulting identity network is equivalent. This greatly simplifies our calculation, since we are allowed to pick appropriate triangulation that is most convenient.

Note that we have not considered triangulation invariance for path-integral on manifolds with arbitary topology, or with conformal boundaries,, which will be left for future work.

In the rest of this section we compute ensemble averaged path-integrals on the two simplest Riemann surfaces, with genus 0 (the sphere) and genus 1 (the torus). Again it can be explicitly shown that their results are independent of triangulation. We will also comment on how to interpret those results from the perspective of gravitational dual.

### 4.1.1 Genus 0

As the simplest example, consider a closed sphere. Picking the simplest triangulation, and applying (4.4), we have

$$\overline{Z_{S^2}} = \frac{1}{g_\alpha g_\beta g_\gamma} \sum_{a,b,c} \left( \text{diagram} \right) = \frac{1}{g_\alpha g_\beta g_\gamma} \sum_{a,b,c} \left( \text{diagram} \right), \qquad (4.22)$$

where the descendants indices are implicit. Following the discussion above, the leading contribution to the ensemble average gives

$$\overline{Z_{S^2}} = \frac{1}{(g_\alpha g_\beta g_\gamma)^2} \sum_{a,b,c} \overline{C_{abc}^{\alpha\beta\gamma} C_{cba}^{\gamma\beta\alpha}} \; \left( \text{diagram} \right)$$

$$\approx \int_0^\infty \left[ \prod_{i\in\{a,b,c\}} dP_i \rho_0(P_i) \right] C_0(P_a, P_b, P_c) \; \left( \text{diagram} \right) \qquad (4.23)$$

$$= \int_0^\infty dP_b dP_a \rho_0(P_a)\rho_0(P_b) \; \left( \text{diagram} \right) = \int_0^\infty dP_b \rho_0(P_b) \left( \text{diagram} \right),$$

where we take the small hole limit for the hole labeled by $\beta$ and recover the actual BCO insertions (the blue line) instead of using the conformal block. The integration of $P_b$ actually gives the path integral of the cylinder, with two boundary conditions $\alpha, \gamma$, i.e.,

$$\overline{Z_{S^2}} \approx \frac{1}{g_\alpha g_\gamma} \int_0^\infty dP_b \left[ g_\alpha g_\gamma \rho_0(P_b) \right] \vcenter{\hbox{}} = \frac{1}{g_\alpha g_\gamma} Z_{\text{cyl}(\alpha\gamma)}(\tau) , \qquad (4.24)$$

where $Z_{\text{cyl}(\alpha\gamma)}$ represents the path integral along the cylinder with circumference $\tau$ (the size of the hole) with two boundary conditions labeled by $\alpha, \gamma$, as defined in (A.32). We have already argued that under large-$c$ limit, the Ishibashi state can be approximated by the Cardy state normalized by $g_\alpha$, the result exactly matches the known sphere partition function [33, 34]

$$\overline{Z_{S^2}} \approx \vcenter{\hbox{}} = \exp\left( \frac{cl}{12} \right) , \quad l = \log \frac{L\epsilon}{\pi} \sin\left( \frac{2\pi x}{L} \right) . \qquad (4.25)$$

The calculation on the sphere is very simple: after applying the ensemble average, all closed loops become contractible loops, since the sphere has genus zero. It is straightforward to verify that different triangulations ultimately yield results analogous to that in (4.23). A more nontrivial example is the computation of the partition function on the torus.

### 4.1.2 Genus 1

Consider evaluating the ensemble averaged path-integral on a torus with the following triangulation,

$$\overline{Z_{T^2}} = \frac{1}{g_\alpha g_\beta g_\gamma g_\rho} \vcenter{\hbox{}} . \qquad (4.26)$$

The opposite sides of the parallelogram must be identified to form a torus. As will be evident, factors of $g$ from the Cardy states would cancel out in the final result here too.

The leading contributions in the ensemble are given by

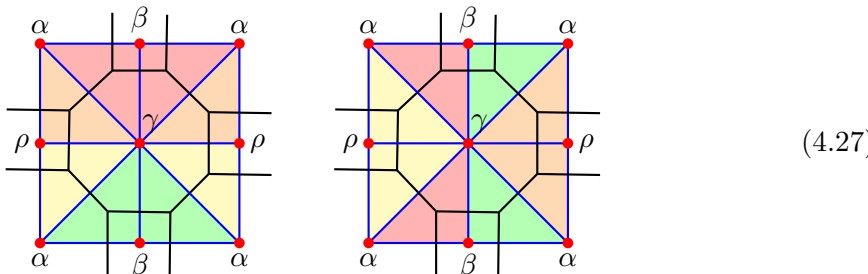

$$(4.27)$$

where triangles of the same color are paired in the Wick contraction, as computed in equation (4.14). Those pairs of triangles form the square block as (4.16). These diagrams lead to the following result,

$$\overline{Z_{T^2}} \approx \int_0^\infty dP_a dP_b \rho_0(P_a)\rho_0(P_b) \left( \vcenter{\hbox{(left diagram)}} + \vcenter{\hbox{(right diagram)}} \right), \quad (4.28)$$

where the purple line delineates the unit cell representing the torus, and the two distinct colored lines correspond to conformal blocks forming closed loops. It is not immediately clear how to read off closed Virasoro characters from this combination of open ones. In the case of RCFT, this issue was solved numerically [2]. To make progress without resorting to numerics, we need some assumptions about the shape and sizes of the triangles. Suppose the two loops in (4.28) are far apart, the descendants of the identity family propagating between the lines are suppressed. To be allowed to make this approximation for loops winding either cycles, it is natural to make the choice $\tau = i$. With this approximation, it appears that the propagation of the two colored lines can be approximated by $\chi_{P_a}(\tau = i)$ and $\chi_{P_b}(\tau = -i)$ respectively. Holomorphic and antiholomorphic blocks for a square unit cell are real and cannot be distinguished. In this approximation, the path-integral could be written as

$$\overline{Z_{T^2}} \approx \int_0^\infty dP_a dP_b \rho_0(P_a)\rho_0(P_b) \left[ \chi_{P_a}(\tau)\chi_{P_b}(\bar{\tau}) + \chi_{P_a}\left(-\frac{1}{\tau}\right) \chi_{P_b}\left(-\frac{1}{\bar{\tau}}\right) \right]$$
$$= |\chi_0(\tau)|^2 + |\chi_0(-1/\tau)|^2 . \qquad (4.29)$$

This result admits a natural gravitational interpretation, as it corresponds to the contributions to the torus partition function from the 3D thermal AdS and BTZ black hole saddles with solid torus topology [35].

Now we would like to verify to what extent our results are independent of triangulation.

Suppose we now consider a simpler triangulation, with the same moduli $\tau = i$,

$$\overline{Z_{T^2}} = \frac{1}{g_\alpha g_\beta} \sum_{\substack{a,b,c \\ d,e,f}} a \quad \text{[diagram]} \quad a = \frac{1}{g_\alpha^3 g_\beta^5} \sum_{\substack{a,b,c \\ d,e,f}} \overline{C_{adc}^{\beta\alpha\alpha} C_{bcf}^{\beta\alpha\alpha} C_{bed}^{\beta\alpha\alpha} C_{afe}^{\beta\alpha\alpha}} \quad a \quad \text{[diagram]} \quad a$$

(4.30)

Applying Wick contraction

$$\overline{C_{adc}^{\beta\alpha\alpha} C_{bcf}^{\beta\alpha\alpha} C_{bed}^{\beta\alpha\alpha} C_{afe}^{\beta\alpha\alpha}} \approx \mathbf{C}_{adc}\mathbf{C}_{aed}\delta_{df}\delta_{ab} + \mathbf{C}_{adc}\mathbf{C}_{edb}\delta_{ec}\delta_{ab} + \sqrt{g_\alpha g_\beta}\mathbf{C}_{adc}\delta_{cf}\delta_{ed}\delta_{b\mathbb{1}} + \sqrt{g_\alpha g_\beta}\mathbf{C}_{bcf}\delta_{dc}\delta_{ef}\delta_{a\mathbb{1}},$$

(4.31)

we could then simplify (4.30) to

$$\int_0^\infty \rho_0(P_a) dP_a \left[ a \quad \text{[diagram]} + a \quad \text{[diagram]} + a \quad \text{[diagram]} + \quad \text{[diagram]} \right].$$

(4.32)

At first glance this result appears different from the one we have already obtained in (4.28). Nevertheless, it can be verified that the topological charge of those results are the same by a further series of crossing transformations in (4.28). Meanwhile, consider a triangulation with more triangles. Using the same set of collection of contracting small loops and a series of crossing relation, the result reduces to an identity network that winds around the A-cycle and B-cycle of the torus, which is the same as the result with very few triangles. Crossing transformations to a specific triangulation should produce the same result. But matching that result to closed Virasoro characters because it involves translating the geometric information of the triangle to the moduli of manifold, which is difficult in general. Only in very special choice of triangulation, such as the one in leading to (4.28), is the conversion to closed conformal blocks appearing to be possible.

### 4.1.3   Diagonal closed CFT from open CFT data

In the above we made extra restrictions to the shape of triangles and the modulus of the torus, to argue that closed Virasoro blocks can be read off more readily in this limit. We would like to supply extra evidence that this can be done more generally. The discrete formulation was initially formulated for rational CFTs. Therefore if anything we should expect that such tricks can be applied to the discrete torus partition function and recover known results about their torus partition functions on the torus. Let us illustrate this with diagonal rational CFTs as an explicit example. We first use four triangles to discretize the

torus partition function as we did above, and get[9]

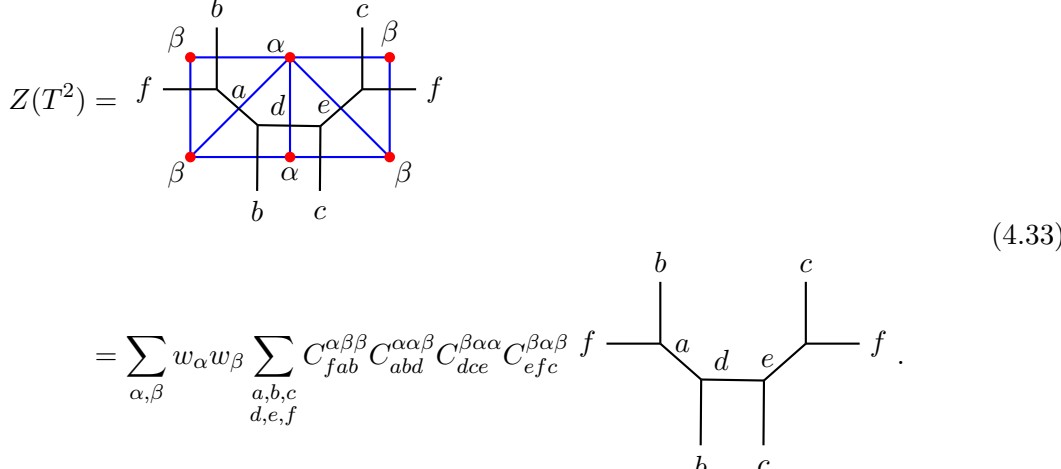

$$(4.33)$$

$$= \sum_{\alpha,\beta} w_\alpha w_\beta \sum_{\substack{a,b,c \\ d,e,f}} C_{fab}^{\alpha\beta\beta} C_{abd}^{\alpha\alpha\beta} C_{dce}^{\beta\alpha\alpha} C_{efc}^{\beta\alpha\beta} \; f$$

For simplicity, we shifted our triangulation comparing to earlier sections, and focus on purely imaginary modular parameter $\tau$ in this section. As introduced in Appendix B, for rational CFT, the weight is given

$$w_\alpha = \mathbb{S}_{\mathbb{1}\mathbb{1}}^{1/2} \mathbb{S}_{\mathbb{1}\alpha} = \mathbb{S}_{\mathbb{1}\mathbb{1}}^{3/2} d_\alpha, \quad d_\alpha := \frac{\mathbb{S}_{\mathbb{1}\alpha}}{\mathbb{S}_{\mathbb{1}\mathbb{1}}}, \tag{4.34}$$

where $d_\alpha$ is the so-called quantum dimension. We first exchange the $b, f$ and $c, f$ legs, introducing phase factors arising from braiding [25, 26],

$$Z(T^2) = \sum_{\alpha,\beta} w_\alpha w_\beta \sum_{\substack{a,b,c \\ d,e,f}} e^{-i\pi(h_b+h_f-h_a)} e^{-i\pi(h_c+h_f-h_e)} C_{fab}^{\alpha\beta\beta} C_{abd}^{\alpha\alpha\beta} C_{dce}^{\beta\alpha\alpha} C_{efc}^{\beta\alpha\beta}$$

$$(4.35)$$

Then we perform three crossing moves turning the conformal blocks into,

$$= \sum_{g,h,p} \mathbb{F}_{P_a P_g} \begin{bmatrix} P_f & P_d \\ P_b & P_b \end{bmatrix} \mathbb{F}_{P_e P_h} \begin{bmatrix} P_d & P_f \\ P_c & P_c \end{bmatrix} \mathbb{F}_{P_d P_p} \begin{bmatrix} P_f & P_f \\ P_g & P_h \end{bmatrix}$$

$$(4.36)$$

---

[9]Here, we adopt a slightly different convention from the one used earlier: $C$ denotes the normalized three-point structure coefficients, following the convention in [1, 2], where they take the simplified form shown in (4.38) when expressed in the Racah gauge.

Consider the kinematical regime where we have a very fat torus, and a small $f$ bubble giving a contribution proportional to the quantum dimension $d_f$. Up to some overall constant normalization factors, the answer from the triangulation becomes,

$$
Z(T^2) = \sum_{\alpha,\beta} d_\alpha d_\beta d_f \sum_{\substack{a,b,c \\ e,f}} e^{-i\pi(h_b+h_f-h_a)} e^{-i\pi(h_c+h_f-h_e)} C_{fab}^{\alpha\beta\beta} C_{abd}^{\alpha\alpha\beta} C_{dce}^{\beta\alpha\alpha} C_{efc}^{\beta\alpha\beta}
$$
$$
\mathbb{F}_{P_a \mathbb{1}} \begin{bmatrix} P_b & P_b \\ P_f & P_f \end{bmatrix} \mathbb{F}_{P_e \mathbb{1}} \begin{bmatrix} P_f & P_f \\ P_c & P_c \end{bmatrix} \chi_b(\tau) \bar{\chi}_c(\bar{\tau})
$$

(4.37)

For diagonal rational CFTs, we can go to the "Racah gauge" [1, 36], where the BCFT structure coefficients are related to quantum $6j$ symbols as,

$$
C_{ijk}^{abc} = (d_i d_j d_k)^{1/4} \begin{Bmatrix} P_i & P_j & P_k \\ P_a & P_b & P_c \end{Bmatrix}_{\mathrm{R}}.
$$

(4.38)

The crossing kernel can be also written as

$$
\mathbb{F}_{P_s P_t} \begin{bmatrix} P_i & P_j \\ P_k & P_l \end{bmatrix} = \sqrt{d_s d_t} \begin{Bmatrix} P_i & P_j & P_t \\ P_l & P_k & P_s \end{Bmatrix}_{\mathrm{R}}, \qquad \mathbb{F}_{P_j \mathbb{1}} \begin{bmatrix} P_i & P_i \\ P_k & P_k \end{bmatrix} = \sqrt{\frac{d_j}{d_i d_k}}.
$$

(4.39)

Combining with the tetrahedral symmetry of the $6j$ symbols

$$
\begin{Bmatrix} P_l & P_m & P_n \\ P_i & P_j & P_k \end{Bmatrix}_{\mathrm{R}} = \begin{Bmatrix} P_m & P_l & P_n \\ P_j & P_i & P_k \end{Bmatrix}_{\mathrm{R}} = \begin{Bmatrix} P_l & P_n & P_m \\ P_i & P_k & P_j \end{Bmatrix}_{\mathrm{R}} = \begin{Bmatrix} P_i & P_j & P_n \\ P_l & P_m & P_k \end{Bmatrix}_{\mathrm{R}},
$$

(4.40)

and the identity associated with the Hexagon identity [25, 26],

$$
\sum_s e^{i\pi h_s} \mathbb{F}_{P_p P_s} \begin{bmatrix} P_j & P_l \\ P_i & P_k \end{bmatrix} \mathbb{F}_{P_s P_q} \begin{bmatrix} P_i & P_l \\ P_k & P_j \end{bmatrix} = e^{-i\pi(h_p+h_q-h_i-h_k-h_j-h_l)} \mathbb{F}_{P_p P_q} \begin{bmatrix} P_i & P_l \\ P_j & P_k \end{bmatrix},
$$

(4.41)

we can perform the sum over $a$ and $e$, and see that the phases are all cancelled, leading to,

$$
\sum_{\substack{\alpha,\beta \\ f,b,c}} d_f \mathbb{F}_{P_\beta P_\alpha} \begin{bmatrix} P_\alpha & P_f \\ P_b & P_\beta \end{bmatrix} \mathbb{F}_{P_\beta P_\alpha} \begin{bmatrix} P_\alpha & P_f \\ P_c & P_\beta \end{bmatrix} \chi_b(\tau) \bar{\chi}_c(\bar{\tau})
$$
$$
= \sum_{\substack{\alpha,\beta \\ f,b,c}} \frac{d_\alpha d_\beta}{\sqrt{d_b d_c}} \mathbb{F}_{P_b P_f} \begin{bmatrix} P_\alpha & P_\alpha \\ P_\beta & P_\beta \end{bmatrix} \mathbb{F}_{P_f P_c} \begin{bmatrix} P_\beta & P_\alpha \\ P_\beta & P_\alpha \end{bmatrix} \chi_b(\tau) \bar{\chi}_c(\bar{\tau})
$$
$$
= \sum_{\substack{\alpha,\beta \\ b,c}} \frac{d_\alpha d_\beta}{\sqrt{d_b d_c}} \delta_{bc} N_{\alpha\beta}^b \chi_b(\tau) \bar{\chi}_c(\bar{\tau})
$$
$$
= \mathcal{D}^2 \sum_b \chi_b(\tau) \bar{\chi}_b(\bar{\tau})
$$

(4.42)

In going to the second equality, we use the orthogonality of the crossing kernels, where $N^b_{\alpha\beta}$ is non-zero only when fusion between the three primaries is allowed. In the final step, we apply the Verlinde formula and the unitarity of the modular $S$-matrix [37]. Here, $\mathcal{D}$ denotes the total quantum dimension.

## 4.2 Path-integral over manifolds with operator insertions and generalized free fields

In our model, we can also study path integrals with closed CFT operator insertions. We will see how the structure of generalized free fields emerge in the loop model coming from the large-$c$ ensemble.

Let us focus on a pair of identical closed operator insertions $O_{h,\bar{h}}$. We choose a triangulation such that both operators lie within the same triangle. This triangle then corresponds to a disk correlation function involving two bulk CFT operators and three BCOs. This correlation function can be computed by performing the bulk-boundary OPE twice, followed by evaluating the resulting BCO five-point correlation function. Diagrammatically, this process is illustrated in the left and middle panels of Fig. 4.

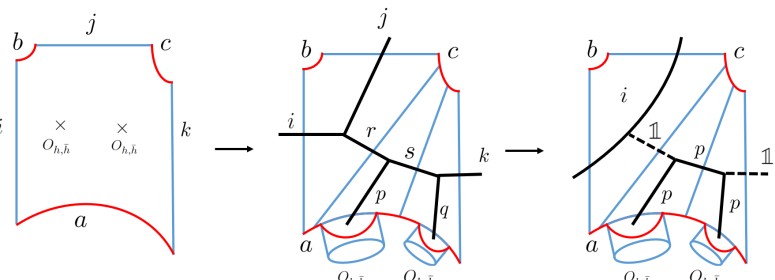

**Figure 4**. The correlation function with two bulk operator insertions in a triangle can be computed by performing the bulk-boundary OPE twice, reducing it to a five-point function of BCOs.

Here, the "whistle" refers to the bulk-to-boundary OPE, where a bulk operator $O_{h,\bar{h}}$ is expanded into boundary operators labeled by $p$ and $q$. For clarity, we depict the whistle outside the triangles in our diagrams.

When we consider triangles not directly connected to the whistles, the ensemble averaging rule remains unchanged and reproduces the loop model structure described above. Now consider averaging over the component connected to the whistles: the average over the bulk-to-boundary OPE coefficients effectively identifies the $p$ and $q$ legs, as in (2.38). Combining these observations, the dominant contribution is given by the right-hand side

of Fig. 4. Plugging in the formulas for the averaged OPE coefficients, this gives

$$\sum_P |C^a_{(h,\bar h),P}|^2 \ \ \overset{P_h\ \ \mathbb{1}\ \ \ \ \mathbb{1}\ \ P_h}{\underset{P_{\bar h}\ \ \ \ P\ \ \ \ P_{\bar h}}{\rangle\!\!\!\!\langle}} \ \ \to \int dP \rho_0(P) C_0(P_h, P_{\bar h}, P) \ \ \overset{P_h\ \ \mathbb{1}\ \ \ \ \mathbb{1}\ \ P_h}{\underset{P_{\bar h}\ \ \ \ P\ \ \ \ P_{\bar h}}{\rangle\!\!\!\!\langle}}$$

$$= \ \ \overset{\mathbb{1}\ \ \ \ \ \ \mathbb{1}}{\underset{}{}} P_h \ \underline{\quad\quad} \ P_h \quad\quad \mathbb{1}$$

$$P_{\bar h} \ \underline{\quad\quad} \ P_{\bar h}$$

(4.43)

This means the final result arises by first fusing the two bulk operators into the bulk identity operator, and then projecting onto the identity on the boundary. Essentially, this indicates that the averaged effect of the bulk-to-boundary OPE behaves as if the boundaries were absent, and simply reproduces the structure of a two point function.

More generally, as discussed near (4.15), ensemble average of any nearest neighbour pair of triangles produce parallel line segments crossing through the region. i.e. there is a form of Gauss law, and non-trivial lines cannot end in the middle of nowhere. In the presence of the insertion of an extra triangle connected to the whistle, a line can end in the whistle. Therefore whistle plays the role of a source of these block lines. Therefore, the ensemble average of such configurations would produce a line connecting one whistle insertion to another, in a background of loops, essentially by Gauss's law. These lines are open CFT propagators, and thus naturally lines of longer physical length are relatively suppressed. Therefore, in complete generality, the leading contribution of these two point functions are controlled by the shortest path connecting these whistles in a background of closed loops.

We can even consider insertion of more whistles. Since lines can only end on whistles and they cannot either end or cross in the bulk to leading order in the ensemble average, these correlation functions would admit a leading contribution of lines joining pairs of whistles, as if they were computed by Wick contraction of pairs of insertions. The schematic diagram is shown in Figure 5. The correlation function of these closed CFT operators thus admit an interpretation as a generalised free field in the ensemble average.

## 5  Conclusions

In this paper, starting from the asymptotic behavior of Boundary Conformal Field Theory (BCFT), we extend the idea of ensemble averaging from bulk Conformal Field Theory (CFT) to BCFTs, constructing the ensemble by treating the OPE coefficients as random variables. We generalize the constructions of [8, 9, 12, 13] to BCFT data, and propose ensemble averages for both BCO structure coefficients and open-closed coupling constants. We discuss in detail how four-point crossing symmetry is preserved by (2.27), and how higher-point crossing symmetry requires the introduction of non-Gaussianities, similar to the situation in closed CFT ensemble averages.

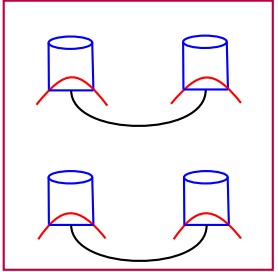 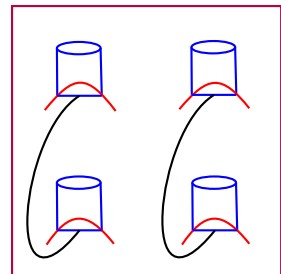 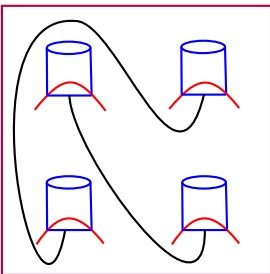

**Figure 5**. Schematic diagram for inserting four whistles. These three are possible diagrams contributing to the averaged correlation functions. Each consists of pairs of whistles connected by lines. As explained these lines cannot intersect or end in the bulk, thus leading to correlation functions that take the form of wick contracting pairs of insertions.

We computed multi-copy observables in the BCFT ensemble, focusing in particular on multi-copy correlation functions. Since a BCFT preserves only a single copy of the Virasoro algebra, it is natural to expect that all results in BCFTs should correspond to the "square root" of those in closed CFTs. Based on the AdS/CFT duality, we anticipate that these results should match those of the Virasoro TQFT, whose partition function is roughly the "square root" of the bulk gravity partition function. Indeed, we observed this correspondence, confirming the consistency of our approach and providing deeper insight into the relationship between boundary and bulk theories.

In prior works, a novel discretization method for CFTs on a surface was proposed, utilizing a triangulation of the CFT path integral in terms of three-point functions of BCOs. The path integral is thereby expressed as a product of open structure coefficients of BCOs, which naturally motivates considering the ensemble average of these path integrals in the large-$c$ limit. Applying the ensemble average we propose, the path integral reduces to a sum over loops. By analyzing the volume of phase spaces that contribute to the ensemble average, we identify the leading-order diagrams—those with the maximum number of loops—as the dominant contributions. Importantly, we demonstrate that our results are independent of the choice of triangulation. From these diagrams, we extract the asymptotic behavior of the path integral in the large-$c$ limit, showing that it is independent of the specific choice of CFT, suggesting a universal structure. While there are subtleties in reading off closed CFT conformal blocks from open ones, by considering special geometries, such a comparison becomes possible. In these limits, we present strong evidence that the torus path integral matches the result obtained from the solid torus computation in pure three-dimensional gravity.

The ensemble average of the BCFT thus appears to provide a more microscopic perspective on the emergence of gravity. It would be interesting to investigate subleading contributions to the ensemble averages. We expect that non-locality would begin to emerge, as triangles that are spatially farther apart in the triangulated path integral should also contribute through Wick contractions in the ensemble average. These are important issues

that we hope to explore in future work.

## Acknowledgments

We thank Lin Chen for initial collaboration on the project. We thank Diandian Wang for helpful comments on the draft. BL thanks Herman Verlinde and Mengyang Zhang for helpful advice. Part of this work was done during the workshop "IAS Quantum Information and Physics Focus Week Workshops" in IAS, and also the KITP Program "Generalized Symmetries in Quantum Field Theory: High Energy Physics, Condensed Matter, and Quantum Gravity" in KITP. This research was supported in part by grant NSF PHY-2309135 to the Kavli Institute for Theoretical Physics (KITP). The work of YJ is supported by the U.S Department of Energy ASCR EXPRESS grant, Novel Quantum Algorithms from Fast Classical Transforms, and Northeastern University. LYH acknowledges the support of NSFC (Grant No. 11922502, 11875111).

## A   Review and conventions for CFT and BCFT structure coefficients

### A.1   Bulk operators structure coefficients

To set notations, we denote bulk operators of a CFT inserted at point $z$ on the complex plane by $\mathcal{O}_i(z)$, where $i$ is the label of the primary. By the state-operator correspondence, these operators correspond to a state $|\mathcal{O}_i(z)\rangle$ defined on a closed circle $S^1$. The Hilbert space of states, denoted $\mathcal{H}_{\text{closed}}$, is given by

$$\mathcal{H}_{\text{closed}} = \oplus_{i \in \mathcal{S} \times \bar{\mathcal{S}}} \mathcal{M}_{i,\bar{i}} \mathcal{V}_i \otimes \overline{\mathcal{V}}_{\bar{i}}, \tag{A.1}$$

where $\mathcal{S} \times \bar{\mathcal{S}}$ means the spectrum of the CFT on the circle, $\mathcal{M}_{i,\bar{i}}$ is the multiplicity and $\mathcal{V}_i$ ($\overline{\mathcal{V}}_{\bar{i}}$) is the irreducible representation of holomorphic (antiholomorphic) Virasoro algebra labeled by $i$ ($\bar{i}$). The bulk operators have conformal dimension $h_i$ and $\bar{h}_{\bar{i}}$, with scaling dimension $\Delta_i = (h_i + \bar{h}_{\bar{i}})/2$ and spin $J_i = (h_i - \bar{h}_{\bar{i}})/2$.

Consider a bulk operator $\mathcal{O}_i$ with conformal dimension $(h_i, \bar{h}_i)$. The two point functions and three point functions in the complex plane are completely fixed by conformal symmetry. The three point functions for primary operators $\mathcal{O}_{i,j,k}$ is of the following form

$$\langle \mathcal{O}_i(z_1, \bar{z}_1) \mathcal{O}_j(z_2, \bar{z}_2) \mathcal{O}_k(z_3, \bar{z}_3) \rangle = \frac{C_{ijk}}{z_{ij}^{h_{ij,k}} z_{jk}^{h_{jk,i}} z_{ki}^{h_{ki,j}} \times \text{(anti-holomorphic)}}, \tag{A.2}$$

where $z_{ij} = z_i - z_j$, $h_{ij,k} = h_i + h_j - h_k$. Here $C_{ijk}$ is the three point structure coefficient of the bulk operator, satisfying

$$C_{ijk} = C_{jki} = (-1)^{J_i + J_j + J_k} C_{jik}, \tag{A.3}$$

where the first equal holds if the spins of the operator are integers or half integers. Also one can consider $C_{ijk}^*$ and find $C_{ijk}^* = (-1)^{J_i + J_j + J_k} C_{ijk} = C_{kji}$. Note that we have implicitly

used the normalization that the two point correlation function for primary operators $\mathcal{O}_{i,j}$ on the sphere is

$$\langle \mathcal{O}_i(z_1, \bar{z}_1) \mathcal{O}_j(z_2, \bar{z}_2) \rangle_{S^2} = \frac{\delta_{ij}}{z_{12}^{2h_i} \bar{z}_{12}^{2\bar{h}_i}} . \tag{A.4}$$

Correlation functions involving descendants can be obtained from those of primaries using conformal symmetries, and they share the same structure coefficients with the primaries. We will therefore only discuss primaries explicitly. Another important quantity is the four point correlation function

$$\langle \mathcal{O}_1(z_1, \bar{z}_1) \mathcal{O}_2(z_2, \bar{z}_2) \mathcal{O}_3(z_3, \bar{z}_3) \mathcal{O}_4(z_4, \bar{z}_4) \rangle = \prod_{i<j} z_{ij}^{h/3 - h_i - h_j} \prod_{i<j} \bar{z}_{ij}^{\bar{h}/3 - \bar{h}_i - \bar{h}_j} G(x, \bar{x}), \tag{A.5}$$

where $h = \sum_i h_i$, $\bar{h} = \sum_i \bar{h}_i$ and $x$ ($\bar{x}$) is the holomorphic (anti-holomorphic) cross ratio $x := \frac{z_{12}z_{34}}{z_{13}z_{24}}$ ($\bar{x} := \frac{\bar{z}_{12}\bar{z}_{34}}{\bar{z}_{13}\bar{z}_{24}}$). The function $G(x, \bar{x})$ can be expressed in terms of the conformal blocks

$$G_{1234}(x, \bar{x}) = \sum_m C_{12m} C_{34m} \mathcal{F}^s_{1234}(m|x) \overline{\mathcal{F}}^s_{1234}(m|\bar{x}), \tag{A.6}$$

where $\mathcal{F}^s_{1234}(m|x)$ is called the $s$-channel Virasoro conformal block depending on the conformal dimensions $h_{1,2,3,4,m}$ and the cross ratio $x$ (similar for the anti-holomorphic part $\overline{\mathcal{F}}^s_{1234}(m|x)$). For convenience, we adopt the notation representing the conformal blocks in the following way

$$\tag{A.7}$$

and define

$$\tag{A.8}$$

From now on, when we use the notation $|A|^2$, it denotes the product of $A$ (the holomorphic part) and its anti-holomorphic part $\bar{A}$, i.e., $|A|^2 := A\bar{A}$. $G(x, \bar{x})$ can be also computed via expansion in a different channel

$$G(x, \bar{x}) = \sum_q C_{41n} C_{23n} \mathcal{F}^t_{1234}(n|1-x) \overline{\mathcal{F}}^t_{1234}(n|1-\bar{x}), \quad \mathcal{F}^t_{1234}(n|1-x) = \quad$$

$$\tag{A.9}$$

where $\mathcal{F}^t_{1234}(m|1-x)$ is called the $t$-channel Virasoro conformal block and $\mathcal{F}^t$ is related to

$\mathcal{F}^s$ by the crossing kernel

$$\sum_{t} \mathbb{F}_{st} \begin{bmatrix} 1 & 4 \\ 2 & 3 \end{bmatrix} \qquad (A.10)$$

**Some useful identities involving the crossing kernel**

The crossing kernel $\mathbb{F}$ satisfies a series of consistency equations.[10] The crossing kernel is the same when we permute the external indices in the following way

$$\mathbb{F}_{P_s P_t} \begin{bmatrix} P_1 & P_4 \\ P_2 & P_3 \end{bmatrix} = \mathbb{F}_{P_s P_t} \begin{bmatrix} P_4 & P_1 \\ P_3 & P_2 \end{bmatrix} = \mathbb{F}_{P_s P_t} \begin{bmatrix} P_2 & P_3 \\ P_1 & P_4 \end{bmatrix} \qquad (A.11)$$

When $P_m \to \mathbb{1}, P_1 = P_2, P_3 = P_4$, we have

$$\mathbb{F}_{\mathbb{1},P_n} \begin{bmatrix} P_1 & P_3 \\ P_1 & P_3 \end{bmatrix} = C_0\left(P_1, P_3, P_n\right) \rho_0(P_n), \qquad (A.12)$$

where $C_0(P_i, P_j, P_k)$ is

$$C_0(P_1, P_2, P_3) = \frac{\Gamma_b(2Q)\Gamma_b\left(\frac{Q}{2} \pm iP_1 \pm iP_2 \pm iP_3\right)}{\sqrt{2}\Gamma_b(Q)^3 \prod_{j=1}^{3} \Gamma_b\left(Q \pm 2iP_j\right)}, \qquad (A.13)$$

where $\Gamma_b(x)$ is the double Gamma function and $\Gamma_b(x\pm y) := \Gamma_b(x+y)\Gamma_b(x-y)$. $C_0(P_1, P_2, P_3)$ actually relates to the DOZZ formula of Liouville theory, by scaling the operator of $\rho_0(P)$

$$C_0(P_1, P_2, P_3) = \frac{C_{\text{DOZZ}}(P_1, P_2, P_3)}{\sqrt{\prod_{j=1}^{3} \rho_0(P_j)}}. \qquad (A.14)$$

If there is no ambiguity we often use the abbreviation $\mathbf{C}_{123}$ as defined in (2.37). It is convenient to rewrite the crossing kernel in terms of $6j$ symbol. Using Racah-Wigner normalization we could write

$$\mathbb{F}_{P_s P_t} \begin{bmatrix} P_1 & P_4 \\ P_2 & P_3 \end{bmatrix} = \rho_0(P_t)\sqrt{\frac{C_0(P_2, P_3, P_t)C_0(P_1, P_4, P_t)}{C_0(P_1, P_2, P_s)C_0(P_3, P_4, P_s)}} \begin{Bmatrix} P_1 & P_2 & P_s \\ P_3 & P_4 & P_t \end{Bmatrix}. \qquad (A.15)$$

---

[10]See the recent review in [38]

Geometrically, the $6j$ symbol corresponds to a tetrahedra

$$\begin{Bmatrix} P_1 & P_2 & P_s \\ P_3 & P_4 & P_t \end{Bmatrix} \quad \longleftrightarrow \quad \text{} \quad , \tag{A.16}$$

and the symmetry mentioned in (A.12) can also be interpreted as the tetrahedral symmetry.

The crossing equation for $c > 1$ conformal blocks takes the following form that involves a continuous integral

$$\text{} = \int_0^\infty dP_t \mathbb{F}_{P_s P_t} \begin{bmatrix} P_1 & P_4 \\ P_2 & P_3 \end{bmatrix} \text{} \,. \tag{A.17}$$

**Modular transform and torus Virasoro blocks**

Modular transformation of torus one-point block is implemented by the modular matrix $\mathbb{S}_{P_1,P_2}[P_0]$

$$\text{} = \int_0^\infty dP_2 \mathbb{S}_{P_1,P_2}[P_0] \text{} \,. \tag{A.18}$$

If $P_0 = iQ/2$, the conformal block reduces to the Virasoro character and we have

$$\chi_{P_1}\left(-\frac{1}{\tau}\right) = \int_0^\infty dP_2 \mathbb{S}_{P_1,P_2}[\mathbb{1}] \chi_{P_2}(\tau), \tag{A.19}$$

with

$$\mathbb{S}_{P_1,P_2}[\mathbb{1}] = 2\sqrt{2}\cos\left(4\pi P_1 P_2\right) \quad P_1, P_2 \neq \mathbb{1}\,. \tag{A.20}$$

In addition, if $P_1 = iQ/2$ labels the vacuum, we usually denote the corresponding Virasoro character as $\chi_{\mathbb{1}}(\tau)$ and we have the following equality for the identity block

$$\chi_{\mathbb{1}}\left(-\frac{1}{\tau}\right) = \int_0^\infty dP \mathbb{S}_{\mathbb{1},P}[\mathbb{1}] \chi_P(\tau)\,, \tag{A.21}$$

and

$$\mathbb{S}_{\mathbb{1},P}[\mathbb{1}] = 4\sqrt{2}\sinh\left(2\pi bP\right)\sinh\left(\frac{2\pi P}{b}\right)\,. \tag{A.22}$$

## A.2 Boundary CFT

Now consider introducing conformal boundaries. Consider bulk operator inserted in the upper-half plane (which is conformally related to a disk) with conformal boundary $\alpha$ imposed on the real line. Such a one point function is generically non-zero. The coefficient $\mathcal{B}_\alpha^i$ in (A.31) is referred to as the disk one-point function coefficient, i.e.,

$$\langle \mathcal{O}_i(z) \rangle_\alpha = \frac{\mathcal{B}_\alpha^i}{|z - \bar{z}|^{2h_i}}, \quad \langle \mathbb{1} \rangle_\alpha := g_\alpha = \mathcal{B}_\alpha^{\mathbb{1}}. \tag{A.23}$$

The one point function of the bulk identity operator $g_\alpha$ is actually the disk partition function given the boundary condition $\alpha$, and is sometimes called the $g$-function.

Consider four point correlation function of BCOs

 $= \langle \Psi_1^{\alpha\beta}(x_1) \Psi_2^{\beta\gamma}(x_2) \Psi_3^{\gamma\lambda}(x_3) \Psi_4^{\lambda\alpha}(x_4) \rangle = \prod_{i<j} x_{ij}^{h - h_i - h_j} \hat{G}(\eta),$ (A.24)

where $h = \frac{1}{3}\sum_{i=1}^4 h_i$ and the cross ration is $\eta := \frac{x_{12}x_{34}}{x_{13}x_{24}}$. Notice that the four point correlation function for BCOs takes the form of the holomorphic part in the correlation function of bulk operators. The function $\hat{G}(\eta)$ can be expanded in two distinct channel (around $\eta = 0$ and $\eta = 1$)

$$\hat{G}(\eta) = \sum_{\Psi_s \in \mathcal{H}_{\text{open}}^{\alpha\gamma}} C_{12s}^{\gamma\alpha\beta} C_{s34}^{\lambda\alpha\gamma} \mathcal{F}_{1234}^s(s|\eta) = \sum_{\Psi_s^{\alpha\gamma} \in \mathcal{H}_{\text{open}}^{\alpha\gamma}}$$ ,

$$\hat{G}(\eta) = \sum_{\Psi_t \in \mathcal{H}_{\text{open}}^{\beta\lambda}} C_{41t}^{\beta\lambda\alpha} C_{t23}^{\gamma\lambda\beta} \mathcal{F}_{1234}^t(t|1 - \eta) = \sum_{\Psi_t^{\beta\lambda} \in \mathcal{H}_{\text{open}}^{\beta\lambda}}$$ ,

(A.25)

and they need to be consistent with crossing symmetry, which implies bootstrap constraints on the open structure coefficients. Together with the transformation between the $s$ channel

conformal block and $t$ channel conformal block (A.10), we have

$$\sum_s C_{12s}^{\gamma\alpha\beta} C_{s34}^{\lambda\alpha\gamma} \mathbb{F}_{st} \begin{bmatrix} 1 & 4 \\ 2 & 3 \end{bmatrix} = C_{41t}^{\beta\lambda\alpha} C_{t23}^{\gamma\lambda\beta}. \tag{A.26}$$

**Ishibashi states and Cardy states**

In the following, we will need a few facts about BCFTs i.e. The CFT path-integral is performed on a manifold with boundaries. As explained in [39], consider conformal field theory on the upper half plane $\mathbb{H} := \{z \in \mathbb{C} | \operatorname{Im} z \geq 0\}$ with the real axis as the boundary. conformal invariance is preserved when the stress tensor satisfies the following condition along the real axis

$$T(z)|_{z\in\mathbb{R}} = \bar{T}(\bar{z})|_{z\in\mathbb{R}}. \tag{A.27}$$

At the junction of the boundary $\alpha$ and $\beta$, one can insert the boundary changing operator (BCO) $\Psi_i^{\alpha\beta}(x)$ with conformal dimension $h_i$ with $x$ parametrizing the position along the one dimensional boundary. If the theory lives on the upper half plane with the real axis as the boundary, $x \in \mathbb{R}$. The Hilbert space of states on the boundary is

$$\mathcal{H}_{\text{open}}^{\alpha\beta} = \oplus_{i\in\mathcal{S}^{\alpha\beta}} \mathcal{M}_i^{\alpha\beta} \mathcal{V}_i^{\alpha\beta}, \tag{A.28}$$

where $\mathcal{S}^{\alpha\beta}$ denotes the spectrum of the line with two boundaries $\alpha$ and $\beta$ at the two end points. The Hilbert space of the states on the boundary only contains one copy of Virasoro algebra, which is a direct consequence of the condition (A.27). Mapping the upper half plane with conformal boundary condition $\alpha$ to the disk, the boundary defines the so-called Cardy states $|B_\alpha\rangle$ in $\mathcal{H}_{\text{closed}}$[39], satisfying the following condition obtained from mode expansion of (A.27)

$$\left(L_n - \overline{L}_{-n}\right)|B_\alpha\rangle = 0, \quad n \in \mathbb{Z}. \tag{A.29}$$

Setting $n = 0$ we directly get that the state $|B_\alpha\rangle$ have vanishing spin, belonging to the scalar sector of $\mathcal{H}_{\text{closed}}$, which we denote by $\mathcal{H}_{\text{closed}}^{\text{sc.}}$. We can construct a set of basis states $|B, i\rangle\rangle$ satisfying (A.29). These basis states are called Ishibashi states[40]. They are constructed form each primary family $i$ with vanishing spin. They take the form

$$|B, i\rangle\rangle = \sum_{N=0}^{\infty} \sum_{j=1}^{d(N)} |i, N; j\rangle \otimes \overline{|i, N; j\rangle}. \tag{A.30}$$

The vacuum Ishibashi state introduced in (B.1) is one special case of (A.30).

A local conformal boundary condition corresponds to the boundary state can be written as the linear superposition of Ishibashi states

$$|B_\alpha\rangle = \sum_{i\in\mathcal{H}_{\text{closed}}^{\text{sc.}}} \mathcal{B}_\alpha^i |B, i\rangle\rangle. \tag{A.31}$$

These coefficients $\mathcal{B}_\alpha^i$ are solutions to the Cardy condition [39], which is essentially the open-closed duality on the cylinder.

## Bulk-Boundary correspondence and Cardy's condition

Consider the path integral $Z_{\mathrm{cyl}(\alpha\beta)}$ along the cylinder with circumference $\tau$ and two boundary conditions labeled by $\alpha, \beta$.

$$Z_{\mathrm{cyl}(\alpha\beta)}(\tau) = \qquad\qquad . \tag{A.32}$$

Inserting a complete set of basis states on the interval between the boundary $\alpha, \beta$ we can get

$$Z^{\mathrm{open}}_{\mathrm{cyl}(\alpha\beta)}(\tau) = \qquad = \sum_{\Psi_i^{\alpha\beta}\in\mathcal{H}^{\alpha\beta}_{\mathrm{open}}} n_i^{\alpha\beta}\chi_i(\tau) = \int_0^\infty dP \rho_{\alpha\beta}^{\mathrm{open}}(P)\chi_P(\tau), \tag{A.33}$$

where $n_i^{\alpha\beta}$ are the multiplicities and $\rho_{\alpha\beta}^{\mathrm{open}}(P)$ is the spectrum density. Formally we can write

$$\rho_{\alpha\beta}^{\mathrm{open}}(P) := \sum_{\Psi_i^{\alpha\beta}\in\mathcal{H}^{\alpha\beta}_{\mathrm{open}}} n_i^{\alpha\beta}\delta(P - P_i). \tag{A.34}$$

We can also insert bulk operators in another channel which gives

$$Z^{\mathrm{closed}}_{\mathrm{cyl}(\alpha\beta)}\left(-\frac{1}{\tau}\right) = \qquad\qquad$$

$$= \sum_{\mathcal{O}_i\in\mathcal{H}^{\mathrm{sc.}}_{\mathrm{closed}}} \mathcal{B}_\alpha^i\mathcal{B}_\beta^i\chi_i\left(-\frac{1}{\tau}\right) = \int_0^\infty dP \rho_{\alpha\beta}^{\mathrm{closed}}(P)\chi_P\left(-\frac{1}{\tau}\right), \tag{A.35}$$

where we also formally define

$$\rho_{\alpha\beta}^{\mathrm{closed}}(P) := \sum_{\mathcal{O}_i\in\mathcal{H}^{\mathrm{sc.}}_{\mathrm{closed}}} \mathcal{B}_\alpha^i\mathcal{B}_\beta^i\delta\left(P - P_i\right). \tag{A.36}$$

Cardy's condition is the requirement that the open and closed CFT competition agrees. Using the modular transformation (A.19) between $\chi_P(\tau)$ and $\chi_P(-1/\tau)$, the equivalence of these two channels implies

$$\rho_{\alpha\beta}^{\mathrm{open}}(P) = \int_0^\infty dP' \mathbb{S}_{PP'}[\mathbb{1}]\rho_{\alpha\beta}^{\mathrm{closed}}(P'). \tag{A.37}$$

Substitute the definition of $\rho_{\alpha\beta}^{\text{closed}}(P)$ we have

$$\rho_{\alpha\beta}^{\text{open}}(P) = g_\alpha g_\beta \mathbb{S}_{P\mathbb{1}}[\mathbb{1}] + \sum_{\substack{\mathcal{O}_i \in \mathcal{H}_{\text{closed}}^{\text{sc.}} \\ \mathcal{O}_i \neq \mathbb{1}}} \mathcal{B}_\alpha^i \mathcal{B}_\beta^i \mathbb{S}_{PP_i}[\mathbb{1}] .$$

(A.38)

## A.3 Revisit crossing symmetry

In this part we revisit the crossing symmetry violation problem and carefully examine in what sense the crossing symmetry survives. Notice that the label of boundary condition and the label of inserted operators factorize when we take ensemble average, the crossing symmetry violation problem in BCFT ensemble average is exactly the same as in the CFT ensemble average. To avoid unnecessary complications caused by too many indices, in this part we discuss the problem in the closed CFT.

This non-Gaussianity is necessary to restore crossing symmetry in higher-point function. Consider higher point correlation function, such as 6-point correlations. Given three distinct operators $\mathcal{O}_{1,2,3}$, none of which equals the identity operator, we can consider the 6 point correlation function expanding in the necklace channel

$$G_{123321}^n(z_i, \bar{z}_i) := \sum_{i,j,k} C_{21i} C_{3ij} C_{3jk} C_{2k1} \left| \begin{array}{c} \phantom{x} \end{array} \right|^2 .$$

(A.39)

In the meanwhile we can also expand it in another channel

$$G_{123321}^f(z_i, \bar{z}_i) := \sum_{i,j,k} C_{1ji} C_{2i3} C_{3jk} C_{2k1} \left| \begin{array}{c} \phantom{x} \end{array} \right|^2 .$$

(A.40)

In CFT these two expansions are equivalent due to crossing symmetry. However, it is no longer guaranteed when we consider CFT ensemble. Taking average of the OPE coefficients,

we have

$$\overline{G^n_{123321}(z_i,\bar z_i)} = \sum_{i,j,k} \overline{C_{21i}C_{3ij}C_{3jk}C_{2k1}}\ \left| \vcenter{\hbox{(block diagram: external legs 2,3,3,2; internal legs $i$, $j$, $k$; horizontal line ending in 1 on both sides)}} \right|^2$$

$$= \sum_{i,j,k} \overline{C_{21i}C^*_{21k}}\ \overline{C_{3ij}C^*_{3kj}}\ \left| \vcenter{\hbox{(block diagram: external legs 2,3,3,2; internal legs $i$, $j$, $k$; 1 ... 1)}} \right|^2 \tag{A.41}$$

$$= \left| \int_0^\infty dP_i\, dP_j\, \rho_0(P_i)\rho_0(P_j)\, \mathbf{C}_{12i}\mathbf{C}_{3ij}\ \vcenter{\hbox{(block diagram: external legs 2,3,3,2; internal legs $i$, $j$, $i$; 1 ... 1)}} \right|^2$$

$$= \left| \vcenter{\hbox{(block diagram: legs 3,3 meeting at $\mathbb{1}$, legs 2,2,1,1)}} \right|^2,$$

where we apply the the Wick contraction in the second line and only keep the non zero contributions. The last line uses the crossing kernel of identity block (A.12) again. A similar computation of the correlations functions can be repeated in another channel. However, if we only include the Gaussian variance

$$\overline{G^f_{123321}(z_i,\bar z_i)} = \sum_{i,j,k} \overline{C_{3jk}C_{k12}C^*_{23i}C^*_{ij1}}\ \left| \vcenter{\hbox{(block diagram: legs 2,3 meeting at $i$, legs 3,2; internal legs $j$, $k$; 1 ... 1)}} \right|^2 = 0. \tag{A.42}$$

Crossing symmetry at this level is restored in the presence of the 4-th moment modification (2.25) to the Gaussian variance, which gives

$$\overline{G^f_{123321}(z_i,\bar z_i)} = \left| \int_0^\infty dP_i\, dP_j\, dP_k\, \rho_0(P_i)\rho_0(P_j)\rho_0(P_k)\times \right.$$

$$\left. \mathbb{F}_{P_k,P_i}\begin{bmatrix} P_2 & P_3 \\ P_1 & P_j \end{bmatrix}\mathbf{C}_{12k}\mathbf{C}_{k3j}\rho_0(P_i)^{-1}\ \vcenter{\hbox{(block diagram: legs 2,3 meeting at $i$, legs 3,2; internal legs $j$, $k$; 1 ... 1)}} \right|^2, \tag{A.43}$$

where we write the 4-th moment in terms of the $\mathbb{F}$ block and use the symmetric property of the crossing kernel (A.11) to exchange the labels for convenience. The inclusion of this

$\mathbb{F}$ block enables us to move the diagram back to the original channel, i.e.,

$$\overline{G^f_{123321}} = \left| \int_0^\infty dP_j dP_k \rho_0(P_j)\rho_0(P_k)\mathbf{C}_{12k}\mathbf{C}_{k3j} \;\; {\scriptstyle\begin{matrix} 2 & 3 & 3 & 2 \\ | & | & | & | \end{matrix}} \atop 1 \rule{2cm}{0.4pt} 1 \;\; \right|^2$$

$$= \left| \vcenter{\hbox{(diagram)}} \right|^2 ,$$ (A.44)

It can be seen that including higher-order moments is necessary to maintain crossing symmetry, at least in the sense of averaging. The key idea is that one needs a crossing kernel $\mathbb{F}$ to move the diagram back to its original channel and this modification can only come from higher moment. Moreover, even including 4-th moment, the crossing symmetry is still violated if we consider more complicated diagrams. Let us provide another concrete example here and explain general interesting structures inside. Given external operators $\mathcal{O}_{1,2,3,4}$ and consider the 8 point correlation functions. Expanding it in the necklace channel and taking the average we will have

$$\overline{G^n_{12344321}}(z_i, \bar{z}_i) = \sum_{i,j,k,l,m} \overline{C_{21i}C_{3ij}C_{4jk}C_{4kl}C_{3lm}C_{2m1}} \left| \;\; \vcenter{\hbox{(diagram)}} \;\; \right|^2$$

$$= \sum_{i,j,k,l,m} \overline{C_{21i}C^*_{21m}}\,\overline{C_{3ij}C^*_{3ml}}\,\overline{C_{4jk}C^*_{4lk}} \left| \;\; \vcenter{\hbox{(diagram)}} \;\; \right|^2$$

$$= \left| \int_0^\infty \prod_{a=i,j,k}[dP_a \rho_0(P_a)]\,\mathbf{C}_{21i}\mathbf{C}_{3ij}\mathbf{C}_{4jk} \;\; \vcenter{\hbox{(diagram)}} \;\; \right|^2$$ (A.45)

$$= \left| \vcenter{\hbox{(diagram)}} \right|^2 ,$$

where the dashed line represents the identity $\mathbb{1}$ as noted previously. Again, if we consider the expansion in another channel, which is related by acting two $\mathbb{F}$-move on the necklace

diagram, we have

$$\overline{G^{ff}_{12344321}(z_i, \bar{z}_i)} = \sum_{i,j,k,l,m} \overline{C_{32i}C_{4ij}C_{1kj}C_{4kl}C_{3lm}C_{2m1}} \left| \begin{array}{c} \end{array} \right|^2 . \quad (A.46)$$

If the ensemble only involves non zero variance and 4-th moment which contributes at most one crossing kernel, one can verify that $\overline{G^{ff}_{12344321}(z_i, \bar{z}_i)}$ is still zero and hence the crossing symmetry breaks again. To restore the crossing symmetry one needs to introduce 6-th moment

$$\overline{C_{4kl}C_{3lm}C_{m12}C^*_{23i}C^*_{i4j}C^*_{jk1}}\Big|_c =$$
$$\left| \sqrt{\frac{\mathbf{C}_{4kl}\mathbf{C}_{3lm}\mathbf{C}_{m12}\mathbf{C}_{23i}\mathbf{C}_{i4j}\mathbf{C}_{jk1}}{\mathbf{C}_{1il}}} \begin{Bmatrix} P_4 & P_j & P_i \\ P_1 & P_l & P_k \end{Bmatrix} \begin{Bmatrix} P_2 & P_3 & P_i \\ P_l & P_1 & P_m \end{Bmatrix} \right|^2, \quad (A.47)$$

which also appears in the six boundary wormhole computation in [24]. One can verify that we can arrive at the same result as (A.45) once this higher moment is included. We have two comments in order. First, we explicitly demonstrated that there is no perfect crossing symmetry if we viewed the OPE coefficients as random variables. We can only preserve the crossing symmetry to the restricted level, to a specific number of external operators and specific moments. We need to introduce infinite moments if we want the theory to be truly crossing invariant.

Second, the higher moment can be derived through similar considerations of crossing invariance and it mush have similar form shown in (A.47). In [9], the authors compute higher point correlation function in a specific channel. To reproduce the same results via other channels one needs to include more and more complicated moments. One way to write these higher moments is via checking crossing symmetry in higher point correlation functions like we discuss above. The alternative way is to use the Feynman rules introduced in [24] or the tensor model [13]. One can also easily construct the corresponding tensor model of BCFT ensemble.

## B  Review of the triangulation formulation of 2D CFT path-integrals

In [1–3], the authors propose a rigorous discrete formulation of the path-integrals of rational CFTs and Liouville theory, the latter of which is an important example of irrational CFT. Here we briefly review this construction.

Suppose we have a 1+1 D CFT path-integral $Z_{\mathcal{M}}$ over a Riemann surface $\mathcal{M}$. We can dig a small hole of radius $r$ on the surface $\mathcal{M}$ and pick a conformal boundary condition labeled $\alpha$ on the rim of the hole. This boundary condition admits a description as a closed CFT state $|B_\alpha\rangle$, known as a Cardy state in the dual channel[39]. To remove these holes, we consider a weighted sum over these conformal boundaries so that the dual state reduces

to the vacuum Ishibashi state, namely

$$\sum_\alpha w_\alpha \left| B_\alpha \right\rangle = \left| \mathbb{1} \right\rangle\rangle, \qquad \left| \mathbb{1} \right\rangle\rangle = \sum_{N=0}^{\infty} \sum_{j=1}^{d(N)} \left| \mathbb{1}, N; j \right\rangle \otimes \overline{\left| \mathbb{1}, N; j \right\rangle}. \tag{B.1}$$

where $d(N)$ represents the degeneracy of the $N$-th descendant. The weight $w_\alpha$ can be derived explicitly. Particularly, for diagonal RCFTs

$$w_\alpha = \mathbb{S}_{\mathbb{1}\mathbb{1}}^{1/2} \mathbb{S}_{\mathbb{1}\alpha}, \tag{B.2}$$

where $\mathbb{S}_{\alpha\beta}$ is the modular $\mathbb{S}$ matrix relating characters evaluated on a torus with modulus $\tau$ and $-1/\tau$

$$\chi_\alpha \left( -\frac{1}{\tau} \right) = \sum_\beta \mathbb{S}_{\alpha\beta} \chi_\beta (\tau). \tag{B.3}$$

In Liouville theory [3], this weight is given by the Cardy density of states[22]

$$w_\alpha = \rho_0(P_\alpha) = 4\sqrt{2} \sinh\left( 2\pi b P_\alpha \right) \sinh\left( 2\pi b^{-1} P_\alpha \right), \tag{B.4}$$

and the summation of $\alpha$ will be replaced by a continuous integral due to the fact that Liouville CFT has a continuous spectrum. Here we use Liouville parametrization introduced in section 2. With the tiny holes, the path integral is expressed as follows,

$$Z_{\mathcal{M}} = \sum_{\{\alpha_v\}} \left( \prod_{\alpha \in \{\alpha_v\}} w_\alpha \right) Z_{\{\alpha_v\}}, \tag{B.5}$$

where $Z_{\{\alpha_v\}}$ denotes the path-integral of the CFT in the presence of a collection of holes labeled by boundary conditions $\{\alpha_v\}$. Between two distinct holes with boundary conditions $\alpha$ and $\beta$, we can insert the resolution of the identity using the complete orthonormal set of states $|i, I\rangle$ in the open Hilbert space $\mathcal{H}_{\text{open}}^{\alpha\beta}$ bounded by the two boundaries $\alpha, \beta$. The identity is, explicitly,

$$\mathbb{1} = \sum_{\Psi_i^{\alpha\beta} \in \mathcal{H}_{\text{open}}^{\alpha\beta}} \sum_I |i, I\rangle \langle i, I|. \tag{B.6}$$

Note that the states are normalised,

$$\langle i, I | j, J \rangle = \delta_{ij} \delta_{IJ}. \tag{B.7}$$

This insertion of identities on chosen edges connecting two holes lead to a triangulation of the path-integral as shown in figure 6a. The path integral of the CFT is expressed as

$$Z_{\{\alpha_v\}} = \sum_{\{i,I\}} \prod_\Delta Z_{(k,K),(i,I),(j,J)}^{\alpha\beta\gamma}, \quad Z_{\mathcal{M}} = \sum_{\{\alpha_v\},\{i,I\}} \left( \prod_{\alpha \in \{\alpha_v\}} w_\alpha \right) \prod_\Delta Z_{(k,K),(i,I),(j,J),(k,K)}^{\alpha\beta\gamma} \cdot \tag{B.8}$$

The factor $Z_{(k,K),(i,I),(j,J)}^{\alpha\beta\gamma}$ corresponds to the path-integral on a triangle with fixed edge

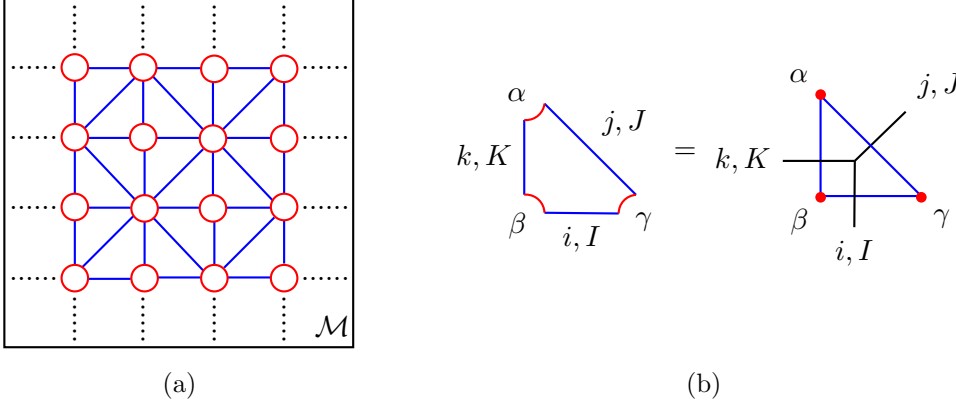

**Figure 6**. (a) Schematic diagram for the triangulation of the two dimensional surface $\mathcal{M}$. The red circle represents the conformal boundary while the blue solid line represents the states we inserted between the boundary. (b) The clipped triangle that forms the triangulation. where $\alpha, \beta, \gamma$ labels the boundary, $i, j, k$ label the primaries and $I, J, K$ represents the descendants. For convenience we usually use the right hand side notation, where we use red dots to represent boundary and also add the dual graph (black solid line) to represent the three point conformal block $\gamma_{ijk}^{IJK}$.

states, and it is conformally related to the three point function of BCOs inserted on the real line in the upper half plane i.e.

$$
Z_{(k,K),(i,I),(j,J)}^{\alpha\beta\gamma} = \quad\;\; \sim \quad\;\; \tag{B.9}
$$

The right hand side denotes the three point function of BCOs in the upper-half plane, with the BCOs inserted at the junctions $0, 1, \infty$ between conformal boundaries. The conformal map governing the shape and size of the triangles is described in detail in [2, 31]. Since these specifics are not central to our discussion, we do not elaborate on them here. Putting these results together, we have

$$
Z_{(k,K),(i,I),(j,J)}^{\alpha\beta\gamma} = \frac{C_{ijk}^{\alpha\beta\gamma}}{\sqrt{g_k^{\alpha\beta} g_i^{\beta\gamma} g_j^{\gamma\alpha}}} \gamma_{ijk}^{IJK} , \tag{B.10}
$$

Here, $C_{ijk}^{\alpha\beta\gamma}$ is the three-point structure coefficient defined in (2.8), $g_k^{\alpha\beta}$ is the two-point normalization factor appearing in (2.7), as we consider *normalized* three-point functions when we insert a complete basis of states, and $\gamma_{ijk}^{IJK}$ encapsulates the three-point conformal

block. Graphically we can define

$$:= \frac{1}{\sqrt{g_\alpha g_\beta g_\gamma}} C_{ijk}^{\alpha\beta\gamma}, \qquad := \gamma_{ijk}^{IJK} . \tag{B.11}$$

Triangulation independence is guaranteed by the crossing symmetry of the BCFT, Pentagon equation and orthogonality relation of 6j symbol. One can use those properties to prove that holes can be closed if the radius of the hole is small, similar to the equation (4.5) we have

$$\sum_\lambda w_\lambda \qquad = \qquad \tag{B.12}$$

During the proof the last step is to prove the hole can be shrunk, as we frequently use in the main text, e.g., the equation (4.9). We need to simplify the following expression

$$\int_0^\infty dP_i \rho_0(P_i) \qquad = \qquad , \tag{B.13}$$

where we use $\tau = \frac{i\epsilon}{2\pi}$ to represent the circumference of the hole. More precisely, suppose the actual radius of the hole on the flat plane is $R$, and the lattice spacing is $L$, then $-\pi/\epsilon = -\ln(L/R)$. The $\epsilon \to 0$ limit corresponds to make the hole infinite small $R \to 0$. On the right hand side the vacuum Ishibashi state is evolved by the closed CFT Hamiltonian $H = L_0 + \bar{L}_0 - c/12$ in the closed channel and the bubble contributes

$$e^{-\frac{2\pi}{\epsilon}\left(L_0 + \bar{L}_0 - \frac{c}{12}\right)} |\mathbb{1}\rangle\rangle = e^{\frac{\pi c}{6\epsilon}} \left( |\mathbb{1}\rangle + \frac{2}{c} e^{-\frac{8\pi}{\epsilon}} L_{-2}\bar{L}_{-2} |\mathbb{1}\rangle + \dots \right) . \tag{B.14}$$

Therefore to recover the CFT path integral, we need to divide every hole by a factor of $e^{\frac{\pi c}{6\epsilon}}$ and we package all these terms into the normalization factor $\mathcal{N}(R)$. The full expression for the path integral is

$$Z_\mathcal{M} = \lim_{R\to 0} \mathcal{N}(R) \sum_{\{\alpha_v\},\{i,I\}} \left( \prod_{\alpha\in\{\alpha_v\}} w_\alpha \right) \prod_\Delta Z_{(k,K),(i,I),(j,J),(k,K)}^{\alpha\beta\gamma} \tag{B.15}$$

Since it does not play any role in our discussion, in the main text we always ignore the normalization factor $\mathcal{N}(R)$ and shrink the hole directly, as how we wrote in (4.9).

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
