# Peer review of "Universal Structures and Emergent Geometry from Large-$c$ BCFT Ensemble"

_SciPost Physics_

## Round 1 · Referee Report · Yuya Kusuki (Referee 1) · 2025-8-13

Disclosure of Generative AI use

The referee discloses that the following generative AI tools have been used in the preparation of this report:

to improve the English

Report

One of the important open problems in the AdS/CFT correspondence is the question: What is the CFT dual to Einstein gravity? A recent development towards this question is the observation that an averaged CFT can describe semiclassical gravity well. Here, an averaged CFT refers to a statistical model in which the OPE coefficients are treated as random variables. This article attempts to extend the concept of averaged CFT to Boundary Conformal Field Theory (BCFT).

First, inspired by the approach of [Chandra–Collier–Hartman–Maloney], the authors introduce a statistical model on manifolds with boundaries by promoting quantities such as the boundary–boundary–boundary OPE coefficients to random variables. Next, they discuss the relation between this “averaged BCFT” and Virasoro TQFT. In contrast to other similar works published around the same time, the authors also investigate the connection to the path integral discretization proposed in their earlier papers. This discretization, as proposed, in principle accommodates manifolds with boundaries. Previously, the authors considered RCFT and Liouville CFT as target theories for this discretization. It is natural to consider the averaged CFT as a third possible target. Motivated by the expectation that discretizing the averaged CFT may provide new insights into AdS/CFT, the authors here attempt to formulate such a discretization. Although many issues remain unresolved in this part, the paper still presents a valuable discussion.

I believe this work is valuable. However, I found several points in the manuscript where the treatment is not thorough. I list the points below and encourage the authors to address them.

Main issues:

  1. Mixing of results valid only in minimal models, general RCFTs, and Liouville CFT. These cases should be clearly distinguished.

1-1. In general RCFT, the Cardy state is not spanned by the Virasoro Ishibashi states but by the Ishibashi states of the maximal chiral algebra. The vacuum Ishibashi states in these two cases are different, but they are conflated throughout this article. Note that this distinction is essential. While one may formally expand RCFT correlation functions in Virasoro blocks for $c>1$, the fusion transformation of the Virasoro block is continuous, which makes it inconsistent with many formulas in this article.

1-2. Results justified only for RCFT are written as if they also apply to Liouville CFT. This seems to refer to results from the authors’ other work. But if RCFT justifications are reviewed, Liouville CFT justifications should also be included; otherwise, readers unfamiliar with the authors’ prior work may be confused.

  1. Equation (4.29) seems to be one of the key results of the paper, but it is not entirely clear to me how nontrivial its verification is. To better understand the precise domain of validity of the approximation, it would be helpful if the authors could describe in more detail—starting from the definition of the conformal block—how the conformal block in Eq. (4.28) is approximated by the character in Eq. (4.29).

Other questions and comments:

  1. In Eq. (2.1), $0<b<1$ is assumed, which implies $c \ge 25$. However, the text says $c>1$. Which regime is actually intended?

  2. The Liouville momentum $\alpha$ lies in ${ \frac{Q}{2} + i\mathbb{R}_+ } \cup [0, \frac{Q}{2} )$. It is generally inappropriate to use the $P$-parameterization for the real domain. For clarity, in places such as below Eq. (2.2) and in Eq. (2.14), it would be better to separate expressions for the real momentum regime from those using $P$.

  3. Solutions to bootstrap equations such as Eq. (2.6) are obtained only in an averaged sense; it might be worth mentioning this explicitly.

  4. In Eq. (2.8), $h_{ij,k}$ is undefined. Also, $x_{ki}$ should presumably be $x_{31}$.

  5. The HKS bound remains unsolved in BCFT; the discussion in Sec. 2.3 could be misleading.

  6. Given that OPE coefficients are generally complex and do not necessarily have a definite sign, I wonder whether it is appropriate to adopt Eq. (2.21) as a general assumption in this work.

  7. The $\gamma$ in Eq. (4.4) is not defined in the main text (it is in the Appendix, but most readers may not notice it).

  8. The authors mention that the triangulation invariance for path-integral on manifolds with arbitrary topology is not considered; what precisely is the obstruction? Also, there is a typo: ,, → ,.

  9. In Eq. (4.29), $\chi(\bar{\tau})$ is likely a typo for $\chi(-\bar{\tau})$. $\overline{\chi(\tau)}$ would also be correct, but note that conjugating $\chi$ also changes the representation to its conjugate.

  10. I would like to see the derivation of the coefficient in Eq. (4.30). At first glance, it appears to yield $g_\alpha^{-5} g_\beta^{-3}$.

  11. The conformal block in Eq. (4.36) is approximated by a character in Eq. (4.37). Could the authors clarify and justify this approximation using the fusion transformation? Under the assumptions stated here, it appears that there could be intermediate states that are not suppressed.

  12. In Eq. (A.1), the definition of the scaling dimension is incorrect.

  13. Both $\bar{h}_{\bar{i}}$ and $\bar{h}_i$ appear; are these intentionally distinguished?

  14. In Eq. (A.2), $z_{ij}$ should be $z_{12}$.

  15. In Eq. (A.8), it seems $m$ is assumed to be a scalar ($\bar{m}=m$). Why?

  16. In Eq. (A.9), the summation index appears incorrect.

  17. The identities in “Some useful identities involving the crossing kernel” are not always valid; assumptions should be stated explicitly.

  18. The definition of the double Gamma function appears to be incorrect.

  19. Typo below Eq. (A.24): ration → ratio.

  20. The definition in Eq. (A.28) is inconsistent with later formulas. The multiplicity should be denoted $n_i^{\alpha\beta}$. Moreover, the superscripts $\alpha\beta$ are unnecessary for the Verma module.

  21. Typo above Eq. (A.30): form → from.

  22. Some symbols in Eq. (A.30) are undefined.

  23. Looking at Eq. (A.35), it appears that $\langle B_\alpha|$ is not defined through BPZ conjugation. Was this choice intentional? (Alternative definitions do exist, but they are typically employed when non-orientable manifolds are part of the discussion.) This subtlety may be irrelevant for specific CFTs, but since the present work addresses a wide variety of CFTs, it may be worth considering.

  24. In the final diagram shown in (A.41), it seems that the contraction should be $11$ rather than $12$. Also, just below (A.41), there appears to be a minor typo: “the the” should read “the”.

  25. In the second equation of Eq. (B.8), the subscript of $Z$ has four indices; is this correct?

Recommendation

Ask for major revision

---

## Round 1 · Referee Report · Anonymous (Referee 2) · 2025-12-13

Report

The paper proposes an ensemble of large-c BCFT data by treating boundary-changing-operator (BCO) structure coefficients (and open–closed couplings) as random variables with Gaussian leading statistics plus non-Gaussian higher moments engineered to restore crossing “order by order.” The ensemble is then applied to averages of (i) multi-copy BCO correlators, claimed to match one copy of Virasoro TQFT. and (ii) triangulated CFT path integrals built from BCO three-point functions, where the ensemble average yields a loop-gas structure and is argued to reproduce features of 3D pure gravity at leading order (with generalized free field behavior for bulk insertions). The direction is interesting, but several core claims currently rely on assumptions and heuristic steps that need substantial tightening. I would require major corrections before the work is reliable.

1.Ensemble definition is not sufficiently well-posed (what exactly is being averaged?) The authors should add a dedicated subsection that formalizes the ensemble as a moment problem (or an explicit effective action / tensor model) and states the domain of validity.

2.The assumption “C _{ijk}^{αβγ}=0” at mean is too hand-wavy. Give a controlled scaling estimate (in the large-c limit and in the relevant operator regimes) for when the mean is negligible relative to the variance contribution, for the specific observables you average (multi-copy correlators; triangulated partition functions). Clarify whether you are assuming an ETH-like “random phase” hypothesis for BCO coefficients analogous to bulk ETH, and state it explicitly.

3.Factorization of boundary labels from operator labels is a very strong assumption. in real BCFT bootstrap data, boundary labels and operator labels are typically entangled (e.g. through fusion rules / classifying algebra / defect data), and this can affect both crossing and the loop-gas mapping. The authors should show at least one nontrivial check (even in a controlled model class) that this factorization reproduces known BCFT constraints beyond a single example channel.

  1. “Crossing restored by higher moments” needs a sharper, more systematic statement. I think authors should specify which set of correlators (4-pt only? 6-pt? arbitrary n-pt?) are crossing-consistent at each truncation of moments.

I would need to recommend acceptance after revision A precise, self-contained definition of the ensemble (variables, symmetry, moments, and their consistency), and a clear statement of what is proven vs assumed. A controlled argument for the loop-gas/adjacent-pairing dominance and triangulation “independence” in the ensemble sense you mean. A more careful (or more modest) gravity comparison, with assumptions spelled out and at least one additional nontrivial check if feasible.

Recommendation

Ask for major revision

---

## Editorial Decision

in_refereeing